# Disturbed Plasma Lipidomic Profiles in Females with Diffuse Large B-Cell Lymphoma: A Pilot Study

**DOI:** 10.3390/cancers15143653

**Published:** 2023-07-18

**Authors:** Romana Masnikosa, David Pirić, Julia Maria Post, Zorica Cvetković, Snježana Petrović, Marija Paunović, Vesna Vučić, Laura Bindila

**Affiliations:** 1Department of Physical Chemistry, Vinca Institute of Nuclear Sciences—National Institute of the Republic of Serbia, University of Belgrade, Mike Petrovica Alasa 12-14, 11000 Belgrade, Serbia; david.piric@vin.bg.ac.rs; 2Clinical Lipidomics Unit, Institute of Physiological Chemistry, University Medical Centre of the J.G.U Mainz, Duesbergweg 6, 55128 Mainz, Germany; 3Department of Haematology, Clinical Hospital Centre Zemun, Vukova 9, 11000 Belgrade, Serbia; 4Faculty of Medicine, University of Belgrade, Dr. Subotića 8, 11000 Belgrade, Serbia; 5Group for Nutritional Biochemistry and Dietology, Centre of Research Excellence in Nutrition and Metabolism, Institute for Medical Research, National Institute of the Republic of Serbia, University of Belgrade, Tadeusa Koscuska 1, 11000 Belgrade, Serbia; snjezana.petrovic@imi.bg.ac.rs (S.P.);

**Keywords:** DLBCL, plasma 4D lipidomics, eicosanoids, HETEs, glycerophospholipids, sphingosine 1-phosphate, sphingolipids, targeted LC-MS, tims-Tof

## Abstract

**Simple Summary:**

Diffuse large B-cell lymphoma (DLBCL) is the predominant type of non-Hodgkin lymphoma—the most common haematological malignancy worldwide. These two cancers are cancers of the immune system and dyslipidemia is a hallmark of both cancer and inflammation. Cancer needs specific lipid species to sustain growth and survival. The global plasma lipidome and sub-lipidome of inflammatory pathways in DLBCL have not yet been reported. In order to address this gap, we conducted targeted lipidomics using liquid chromatography—multiple reaction monitoring to investigate bioactive plasma lipids involved in metabolic and signalling pathways of inflammation and immunity. The global lipidome change in plasma DLBCL was determined by four-dimensional trapped ion mobility mass spectrometry-based lipidomics. In our study, we observed differential lipid profiles in treatment-naïve female patients with DLBCL and disease-free controls. We suggest here the set of sphingosine 1-phosphate, sphingomyelins SM 36:1 and SM 34:1, and phosphatidylinositol PI 34:1 as DLBCL lipid signature, which could serve as a basis for the prospective validation in larger DLBCL cohorts.

**Abstract:**

Lipidome dysregulation is a hallmark of cancer and inflammation. The global plasma lipidome and sub-lipidome of inflammatory pathways have not been reported in diffuse large B-cell lymphoma (DLBCL). In a pilot study of plasma lipid variation in female DLBCL patients and BMI-matched disease-free controls, we performed targeted lipidomics using LC-MRM to quantify lipid mediators of inflammation and immunity, and those known or hypothesised to be involved in cancer progression: sphingolipids, resolvin D1, arachidonic acid (AA)-derived oxylipins, such as hydroxyeicosatetraenoic acids (HETEs) and dihydroxyeicosatrienoic acids, along with their membrane structural precursors. We report on the role of the eicosanoids in the separation of DLBCL from controls, along with lysophosphatidylinositol LPI 20:4, implying notable changes in lipid metabolic and/or signalling pathways, particularly pertaining to AA lipoxygenase pathway and glycerophospholipid remodelling in the cell membrane. We suggest here the set of S1P, SM 36:1, SM 34:1 and PI 34:1 as DLBCL lipid signatures which could serve as a basis for the prospective validation in larger DLBCL cohorts. Additionally, untargeted lipidomics indicates a substantial change in the overall lipid metabolism in DLBCL. The plasma lipid profiling of DLBCL patients helps to better understand the specific lipid dysregulations and pathways in this cancer.

## 1. Introduction

Non-Hodgkin lymphoma (NHL) is the most common haematological malignancy worldwide, ranking as the fifth to ninth most common cancer in most countries [1]. It comprises almost 3% of all cancer diagnoses worldwide, is more common among males, those in their mid-sixties, and those with autoimmune disease or a family history of haematological cancers [2,3]. NHL can occur also in young adults and children [4]. The most frequent NHL subtype in Western countries is diffuse large B-cell lymphoma (DLBCL) comprising 30–40% of all cases [5,6,7]. DLBCL is characterized by the proliferation and accumulation of malignant B cells, both within and outside lymph nodes [3,8]. DLBCL is a genetically, metabolically, phenotypically, and clinically heterogeneous disease, which is usually aggressive and can be divided into molecular subtypes based either on different gene expression profiles or metabolic fingerprints [9,10]. DLBCL resists the efforts to clearly subtype it [11]. The “cell of origin” classification scheme recognises two main DLBCL subtypes: the germinal center B-cell (GCB)-like DLBCL and the activated B-cell (ABC)-like DLBCL, whose cell of origin is less clear. The classification system fails to place 10–15% of cases which remain unclassifiable [3,7,9]. Although DLBCL is nowadays a curable disease, 30–40% of patients are still therapy-resistant or relapse thereafter [12]. Applying modern chemoimmunotherapeutics can help patients achieve long-term control of the disease in nearly 90% of cases given that they had a limited-stage cancer (I and II) at diagnosis; for patients presenting with the advanced stages (III and IV), this is up to 60% [7]. Unfortunately, advanced stage DLBCL accounts for approximately 70% of all cases [3]. Therefore, further research on finding novel biomarkers of DLBCL and revealing molecular mechanisms of the disease are imperatives for improving DLBCL management.

Extensive reprogramming of lipid metabolism emerges as a key factor in cancer development and progression [13,14,15,16,17]. In lymphomas too, metabolic pathways are rewired and lipid metabolism is dysregulated, in processes shaped by oncogenes and tumour-suppressor genes [18]. In some subsets of DLBCL, the dominant energy-generating pathway is the mitochondrial oxidation of palmitate, rather than glycolysis [10,19]. During the course of DLBCL, expectedly abnormal levels of serum lipids were evidenced [20]. Low plasma levels of high-density lipoprotein cholesterol (HDL-C) or its main protein component apolipoprotein A-I (ApoA-I) were found to correlate with an increased risk of several haematological cancers, including NHL [21]. The level of Apo A-I has been recently introduced as a novel prognostic index for DLBCL and combined with the classical International Prognostic Index (IPI) to establish a novel IPI-A risk scoring system [22]. Lipids serve multiple biochemical functions in cancer onset, progression and metastasis [14], hence, the clinical significance of lipid-based biomarkers has been widely recognised [13,23,24,25], driving major undertakings in the global lipidomic profiling of plasma or serum from cancer patients. The majority of cohorts encompassed patients with various solid cancers including breast [26,27], lung [28,29], colorectal [30], prostate [27,31], gastric [29,32], pancreas [33], liver [29], kidney [27], cervical [34], ovarian [35], endometrial [36], thyroid [29] and brain glioma [37]. However, plasma lipidomic studies on haematological cancers are still missing, except for one study on 20 acute myeloid leukaemia (AML) patients [38]. To fill this gap and better understand the extent of the lipidome perturbation in DLBCL, we conducted a pilot study of plasma (sub)lipidome from 17 female DLBCL patients and 21 healthy female controls. The strategy was to maximise, in the same sample cohort, the output of lipid data relevant to advance mechanistic understanding of DLBCL: lipid mediators of immunity and inflammation, i.e., (phospho)sphingolipids and eicosanoids, their upstream precursors, and a more global glycerophospholipidome. For this, we combined multiplexed targeted lipid profiling for investigation of low-abundant signalling lipids with high resolution untargeted lipidomics for broader lipidome coverage.

Inflammation is an overt or covert player in carcinogenesis, where an excessive or unadjusted production of inflammatory molecules invariably occurs [39,40]. DLBCL may also be associated with underlying inflammatory processes, such as those occurring in the DLBCL subset known as host response cluster [19]. Arachidonic acid (AA) and the products of its metabolism, eicosanoids (eiCs), are well-known lipid mediators of pro- and anti-inflammatory pathways [39,41]. Glycerophospholipids (PLs) are sources of AA, where cytosolic (cPLA2) and secretory phospholipase A2 (sPLA2) preferentially release AA from the sn-2 position of a PL molecule [42,43]. AA pathway is interlinked with the metabolism of three other important polyunsaturated fatty acids (PUFAs): linoleic acid (LA), eicosapentaenoic acid (EPA) and docosahexaenoic acid (DHA) [42]. Besides eiCs, other lipid mediators, including lysophospholipids (LPLs), sphingolipids (SLs) such as sphingosine 1-phosphate (S1P) and certain low-abundant PLs such as phosphatidylserine (PS) and phosphatidylinositol (PI) are involved in various signalling pathways [44,45,46,47,48]. Particularly, the role of S1P in pro-cancer signalling events and inflammation has been suggested in various cancers, including DLBCL [47,49,50]. The role played by signalling lipids in the modulation of immune response to cancer attracts an enormous interest, in order to expedite mechanistic elucidation of various cancers. Moreover the general metabolic change of a cancer cell opens new venues for biomarker discovery and cancer phenotyping [51]. An association between DLBCL and conditions affecting the immune system has been established [2]. The intricate role of the immune system during the progression of DLBCL is complex, with myriads of interactions between cancer cells, immunocompetent cells, bioactive/signalling lipid species and structural components from the tumor microenvironment (TME) taking place [52]. Since cancer is inherently linked to alterations in lipid signalling routes [53], we expected that the activation of oncogenic pathway(s) in DLBCL may result in the accumulation of signalling lipids, and, possibly, their precursors. Taking into account all the major lipid classes/subclasses known or suspected to be involved in inflammation and immunity in cancer and the fact that DLBCL is a tumour of immunocompetent cells, we designed and conducted a targeted profiling of selected lipid mediators/signalling lipids, along with their precursors, that is eiCs, resolvin D1 (RvD1), S1P, major SMs and PUFAs, as well as AA-containing PLs. We have also conducted a broad profiling of glycerophospho- and sphingolipidome in plasma of DLBCL patients and controls. The lipidomic portfolio employed in our study included liquid chromatography coupled to multiple reaction monitoring (LC-MRM) and four-dimensional (4D) lipidomics on trapped ion mobility time-of-flight mass spectrometry (tims-Tof MS). 

Herein, we reveal the first plasma lipidomic landscape of DLBCL in a pilot study conducted on female patients. Moreover, we highlight the role of lipid-related inflammatory and immune regulatory processes in this cancer. 

## 2. Materials and Methods

### 2.1. Patients and Control Samples

In our cross-sectional pilot study we enrolled 38 females: 17 DLBCL patients and 21 apparently healthy female controls (CTRL). The study protocol was approved by the Ethics Committee of the Clinical Hospital Centre Zemun, Belgrade, Republic of Serbia, Nr: 145/1 from 21 April 2016. All subjects involved in the study completed a voluntary informed consent to participate in this study. The DLBCL cohort consisted of individuals newly diagnosed with DLBCL, with a median age of 59 years (range 36–69), BMI 19–32, smokers and non-smokers, recruited from the Department of Haematology of Clinical Hospital Centre Zemun, Belgrade, from May 2017. to September 2018. All patients were treatment-naïve. Histological diagnosis of DLBCL was made according to the World Health Organisation [5] after a lymph node biopsy or biopsy of the primary extranodal site. There were eight patients with GCB-like subtype and nine patients with non-GCB subtype of DLBCL. There were no patients with the high-grade B-cell lymphomas in the study. The clinical stage of the cancer was determined by the criteria of the Ann Arbor Conference [54] and individual patients’ risk according to the revised IPI [55]. In order to homogenise the DLBCL cohort, only patients with advanced stages, that is clinical stages III and IV, and predicted poor prognosis according to R-IPI, were included in the present study. Additionally, all patients with previous malignancies, concurrent malignancies, severe chronic diseases, evidence of cancer cachexia or a history of medications/treatments that affect lipid metabolism, such as taking omega-3 supplements or non-steroidal anti-inflammatory drugs (NSAID), were excluded from the study. The CTRL group was recruited from female volunteers working at the University of Belgrade under the following inclusion criteria: no history of malignancy, age over 25 years, BMI range 19–29, on a regular diet, smokers and non-smokers, absence of insulin resistance, type 2 diabetes mellitus or any type of dyslipidemia, autoimmune diseases or systemic inflammation. CTRL females did not take oral contraceptives, omega-3 supplementation or NSAID over three months before the enrolment. The blood from the CTRL females was taken at Clinical Hospital Centre Zemun, Belgrade, in March 2019. 

After an overnight fast, venous blood was taken in the morning into ice-cold EDTA plasma extraction tubes (BD-Plymouth, UK), which were then centrifuged for 15 min at 2000× *g* at 10 °C. The collected plasma was stored at −80 °C until further processing. Prior to the lipid extraction, the plasma was transferred into a new pre-cooled Eppendorf microcentrifuge tube (Eppi) containing previously added 10 µM indomethacin to prevent ex vivo cyclooxygenase (COX)-2 activity. After a short mixing, each plasma sample was aliquoted as following: (i) 20 µL for the untargeted lipidomics, (ii) 100 µL into amber/brown light-protected Eppies for the analysis of eiCs, (iii) 50 µL for the analysis of PLs, SLs and PUFAs, and (iv) the rest of plasma for clinical biochemistry analyses. Plasma was separated within 30 min from the blood withdrawal and the aliquoting was carried out in a cold room. The plasma aliquots were immediately frozen and stored at −80 °C until analysis. 

### 2.2. Plasma Lipid Analysis

#### 2.2.1. Lipid Standards and Materials

ISTDs used for the untargeted lipidomics: PC 17:0_14:1-d5, PE 17:0_14:1-d5, PI 17:0_14:1-d5, Cer d18:1_17:0, SM d18:1_16:1-d9, LPC 17:0-d5, LPE 17:0-d5, LPI 17:1, LPA 17:0, PG 17:0/14:1, were purchased from Merck (Darmstadt, Germany).

Calibration standards used for the targeted lipidomics: AA, thromboxane B2 (TxB2), 5(S)-hydroxyeicosatetraenoic acid (5(S)-HETE), 8(S)-hydroxyeicosatetraenoic acid (8(S)-HETE), 12(S)-hydroxyeicosatetraenoic acid (12(S)-HETE), 15(S)-hydroxyeicosatetraenoic acid (15(S)-HETE), 20-hydroxyeicosatetraenoic acid (20-HETE), 5,6-dihydroxyeicosatrienoic acid (5(6)-DiHET), 8,9-dihydroxyeicosatrienoic acid (8(9)-DiHET), 11,12-dihydroxyeicosatrienoic acid (11(12)-DiHET), 14,15-dihydroxyeicosatrienoic acid (14(15)-DiHET), LA (18:2), EPA (20:5n-3) and DHA (22:6n-3) were purchased from Cayman Chemical (Ann Arbor, MI, USA). RvD1 was obtained from BIOMOL Research Laboratories Inc. (Plymouth Meeting, PA, USA). Lysophosphatidylcholine 18:0/0:0 (LPC 18:0), phosphatidylcholine 16:0/18:1(9Z) (PC 34:1), phosphatidylethanolamine 16:0/18:1(9Z) (PE 34:1), phosphatidylglycerol 16:0/18:1(9Z) (PG 34:1), lysophosphatidic acid 16:0/0:0 (LPA 16:0), phosphatidic acid 16:0/18:1(9Z) (PA 34:1), PS 16:0/18:1(9Z) (PS 34:1), PI 16:0/18:1(9Z) (PI 34:1), SM d18:1/18:0 (SM 36:1), SPH d18:1 and S1P d18:1 were purchased from Avanti Polar Lipids, Inc. (Alabaster, AL, USA). A full list of calibration standards with their target concentration range used in LC-MRM analysis is given in Appendix A.

ISTDs for the targeted lipidomics: AA-d8, TxB2-d4, 5(S)-HETE-d8, 12(S)-HETE-d8, 20-HETE-d6, 11(12)-DiHET-d11, LA-d4, EPA-d5 and DHA-d5 were from Cayman Chemical (Ann Arbor, MI, USA). RvD1-d5 was obtained from BIOMOL Research Laboratories Inc. (Plymouth Meeting, PA, USA). Non-endogeneous lipids used as ISTDs for the quantification of PLs and SLs: PC 17:0/0:0 (LPC 17:0), PA 17:0/0:0 (LPA 17:0), PC 17:0/14:1, PE 17:0/14:1, PG 17:0/14:1, PI 17:0/14:1, PS 17:0/14:1, PA 17:0/14:1, SM d18:1/12:0, S1P d17:1 and SPH d17:1 were from Avanti Polar Lipids, Inc. (Alabaster, AL, USA). The ISTD target concentrations are summarised in Appendix A.

Water, methanol, 2-propanol, acetonitrile (ACN), chloroform, formic acid (FA), ethanol, triethanolamine, and ammonium formate used for the extraction and the LC-MRM analysis were invariably of LC-MS grade (Sigma-Aldrich, St. Louis, MO, USA). LC-grade *tert*-butyl methylether (MTBE), Trizma^®^ hydrochloride solution (Tris-HCl) (pH 7.4) and butyl hydroxytoluene (BHT) were purchased from Sigma-Aldrich. Indomethacin was from Cayman Chemicals. 

#### 2.2.2. Untargeted 4-Dimensional (4D) Lipidomics by Reversed Phase Ultra-High-Pressure LC (UHPLC)-Tims-Tof-MS

A 4D-lipidomic platform using tims-Tof MS was used for untargeted semi-quantitative lipidomic profiling of plasma samples from 17 DLBCL female patients and 21 controls. The robotised lipid extraction, 4D-lipidomics profiling, annotation and data processing were conducted using the protocols, acquisition and processing parameters as recently described [56]. The lipid extraction using MTBE-based liquid-liquid extraction protocol [56,57,58], was carried out from 20 µL plasma. The analysis was performed in negative ion mode. For semi-quantification, peak areas of analytes were normalised to peak areas of class-specific ISTDs, molecular weights and plasma volume, respectively. Lipid levels were then calculated based on spiked amounts of ISTDs and given in nmol/mL.

#### 2.2.3. Targeted Analysis of Plasma Lipids by LC-MRM

Two independent lipid extraction experiments were carried out: eiCs were extracted following the procedure given in [59] whereas PLs, SLs and PUFAs were co-extracted according to the procedures in [59,60,61]. Samples were thawed on ice and 1 µL of the anti-oxidant BHT (100 mM in ethanol) per 1000 µL of plasma was added. All extraction steps were carried out at 4 °C. 

For eiCs extraction, the spike solution contained AA-d8, TxB2-d4, 5(S)-HETE-d8, 12(S)-HETE-d8, 20-HETE-d6, 11(12)-DiHET-d11 and RvD1-d5 (Appendix A). Briefly, 600 μL 0.1 M FA and 50 μL spike solution were added to 100 μL plasma, then vortexed before the addition of 800 µL MTBE. The tubes were mixed for 30 s in Disruptor Genie and immediately placed in a water-ice bath, where a two-phase system was formed. After centrifugation for 15 min at 2000× *g* and 4 °C, the upper layer was transferred to new vials, the solvent was evaporated under a stream of nitrogen at 37 °C and the dried extracts were re-dissolved in 50 µL ACN/0.1 M FA mixture. 20 µL of eiCs extract was injected into the LC-MRM system. For PL/SL/PUFAs co-extraction, the ISTD spike solution contained: PC 14:0_17:1, PE 14:0_17:1, PI 14:0_17:1, PS 14:0_17:1, PG 14:0_17:1, PA 14:0_17:1, SM d18:1_12:0, LPC 17:0, LPA 17:0, LA-d4, EPA-d5, DHA-d5, SPH d17:1; S1P d17:1 (Appendix A). For extraction, 1000 μL MTBE/methanol mixture (10:3; v/v) and 10 μL spike solution were added to 50 μL of plasma samples and vortexed for 1 min at 4 °C. Then, 250 μL 0.1% aqueous FA solution was added, Eppies were mixed for 30 s, cooled on an ice-bath, centrifuged for 10 min (16,000× *g* at 4 °C), the upper organic phase was recovered, evaporated in a nitrogen stream and reconstituted in 90 μL of neat methanol. Half of this was diluted 1:1 (v/v) with methanol, and 10 µL H_2_O was added before injection into LC-MRM. The order of samples was randomised before the lipid extraction, and the samples were randomised before the LC-MRM measurements to exclude any possible bias. 

Targeted quantitative profiling of 35 plasma lipid species in samples from 17 DLBCL female patients and 21 controls was conducted using LC-MRM on a SCIEX 5500 QTrap triple-quadrupole linear ion trap mass spectrometer (AB SCIEX, Darmstadt, Germany), hyphenated to an Agilent 1200 series LC system (Agilent, Waldbronn, Germany) and equipped with a Turbo V Ion Source (AB SCIEX, Darmstadt, Germany) operating in positive/negative ion mode switching [61]. SLs and PLs were analysed in a separate run from PUFAs: LA, AA, EPA and DHA [60,62]. EiCs were quantified in a distinct LC-MRM run as detailed in [59]. Appendix A displays the complete list of quantified lipid species, along with their mean concentrations and units.

### 2.3. Data Analysis

Analyst 1.6.2 (AB SCIEX, Darmstadt, Germany) and MultiQuant 3.0 quantification packages were used for the lipid quantification by LC-MRM and the concentrations of lipids were normalised to the plasma volume. Untargeted data were processed using Metaboscape software version 2021b and an in-house developed 4D library. Processing parameters for 4D data extraction and the criteria for annotation and recursive filters were as reported in [56]. Between-groups comparisons were carried out using Student’s *t*-test, Welch *t*-test or Mann-Whitney U test, after checking the normality (or log_10_-normality) of data distribution (Shapiro-Wilk’s test) and the homogeneity of variance (homoscedasticity) via Bartlet’s or Brown-Forsyth’s test. Means were expressed ± SD and *p*-values were all two-sided. In order to minimise the false discovery (FD), *p*-values derived from multiple comparisons were corrected using the Benjamini-Hochberg procedure (BH). Only variables with BH-corrected *p*-values (q-values) less than 0.05 in the untargeted lipidomics and <0.1 in the targeted lipidomics were considered significant. Statistics was performed by GraphPad Prism v9.4.1., and R packages installed from the free R software environment, https://www.r-project.org (accessed on 1 March 2023). 

For the multivariate data analysis (MDA), all lipid concentrations were converted to pmol/mL. Prior to the principal component analysis (PCA), data were scaled to have zero mean and unit SD. The selection of significant principal components was based on parallel analysis using Monte Carlo simulations. PCA models for the two lipid datasets obtained through the untargeted and targeted lipidomics were built in GraphPad Prism (v. 9.4.1), and re-assessed using SIMCA 17 (version 13.0.3, Umetrics, Umeå, Sweden) and factoextra (v1.0.7) in R. The score plots and loadings were drawn in R. The untargeted lipidomics dataset was further subjected to orthogonal projections to latent structures discriminant analysis (OPLS-DA), after the log_10_-transformation and scaling to the unit variance. The model was validated through the CV-ANOVA. OPLS-DA model was built using R (ropls v1.28.2) and confirmed by SIMCA. 

The lipid dataset from the targeted lipidomics was split for DLBCL samples and controls, and Spearman correlation coefficients (rho, ρ) were computed for each (GraphPad Prism) to create two correlation tables. The significance of the ρ values was assessed through the *p*-values, using an FDR of 5%. Correlation coefficients were used to create heatmaps, whereas statistically significant correlations within groups were used to construct correlation networks, all carried in R. 

## 3. Results

### 3.1. Characteristics of DLBCL Patients and Healthy Controls 

The DLBCL cohort consisted of individuals with a median age of 59 years (range 36–69), BMI 23.8 (19.4–32.2), whereas the CTRL cohort consisted of individuals with a median age of 49 years (range 30–64), BMI 22.6 (19.6–29.5). DLBCL females had levels of triglycerides, total cholesterol, liver enzymes and alkaline phosphatases within normal range (Table 1). 

### 3.2. Plasma Lipidomic Landscape in Females with DLBCL

In order to obtain a more global view of the plasma glycerophospholipidome and sphingolipidome of female patients with DLBCL, in the pilot study presented herein we performed untargeted semi-quantification of 155 lipid species detected in negative ion mode and targeted quantification of 35 lipid species, covering in total eleven lipid classes and twenty subclasses, encompassing eiCs, PUFAs, PLs, LPLs, plasmalogens, ceramides (Cer) and SMs in 17 plasma samples obtained from female DLBCL patients and 21 healthy control females. The semi-quantitative concentrations of lipids in both cohorts are given in Appendix A, alongside the summary of univariate statistics. The mean molar lipid concentrations obtained by the untargeted lipidomics are summarised in Appendix A. 

#### 3.2.1. Overview of Changes in the Plasma Lipidome

A total of 60 lipid molecules from several classes were identified as statistically significantly altered in the plasma of DLBCL female patients compared to the healthy subjects. Only six of these lipid species were elevated in the cancer plasma, that is three Cer species, one LPE species containing docosapentaenoic acid (DPA) and two plasmalogens (ether PLs): PC O-18:1_20:4 and PE O-22:2_20:4, both carrying AA. The remaining 54 lipid species were down-regulated (Table 2). It is worth noting that 22 out of 60 differentially expressed lipids were carriers of LA, predominantly PC species (ten of them), followed by the PE carriers of LA (one PE and five plasmalogens), LPC and LPE (three) and PI carriers of LA (three). These PL species that were significantly down-regulated in DLBCL patients probably mirror an increased mobilisation of LA needed to produce AA, which is in high demand for the cancer cells. Among the highly abundant plasma PC species, a molecule that is a double-carrier of LA, PC 18:2_18:2 is the most significantly diminished lipid species in our untargeted study, with fold change (FC) 0.42. Furthermore, PL species with both LA and AA, that is PC 18:2_20:4, has almost halved in DLBCL plasma. In total, eleven PL species carried AA, among them four PCs, six plasmalogens (five PE O- and one PC O-) and only one PI species. The latter may be due to the efforts to keep some reservoir of PI for the oncogenic signalling. The levels of LPC and LPE carrying LA at sn-1 and sn-2 position, respectively, significantly dropped. Again, it may be that a demanding process of cell membrane remodelling in DLBCL draws this PUFA even from LPLs. Additionally, the LPE may be spent in the signalling. The reservoirs of AA in these PLs also significantly diminished, particularly the species carrying both LA and AA, that is PC 18:2_20:4. The decrease of plasma lipids in DLBCL does not stop at PLs; the plasma pool of SMs is noticeably poorer, with 14 species being significantly affected, mostly SM d41:2 (FC 0.49), SM d32:2 (FC 0.57) and SM d39:1 (FC 0.58). SMs may have been spent to produce the pro-cancer lipid mediator S1P. Four plasma Cer species were differently expressed in DLBCL cohort compared to controls. While Cer d40:2 dropped (FC 0.62), the other three species were increased in DLBCL plasma: Cer d44:1 (FC 1.63), Cer d18:1_24:1 (FC 1.43) and Cer d43:2 (FC 1.58). 

Interestingly, two abundant PI species containing palmitic (16:0) and stearic (18:0) acid, PI 16:0_18:1 and PI 18:0_18:1, were also significantly down-regulated in DLBCL plasma, likely reflecting the augmented consumption of these two PIs in the PI3K signalling cascade. Inspecting Table 2, one cannot but notice the prevalence of unsaturated FAs in all the affected PLs, 21 containing two such FAs in each PL molecule. Finally, only one PS species was differently expressed in DLBCL plasma than in CTRL, PS 18:0_18:1 (PS 36:1), but the change was profound (***). Finally, DHA containing plasma PL species: PE O-17:1_22:6 (FC 0.6), PC 14:0_22:6 (FC 0.66) and PE O-18:2_22:6 (FC 0.76) also showed a lower abundance in the DLBCL cohort, probably being spent for the synthesis of anti-inflammatory lipid mediators, such as RvD1, as a part of coping/compensatory immune response to the cancer state. 

#### 3.2.2. Modelling Plasma Lipidomic Profile in Female Patients with DLBCL

We assessed a possible clustering of plasma samples according to the health state (DLBCL vs. healthy) through visualisation of principal component scores for the two groups. The score plot in Figure 1A shows the distribution of 17 DLBCL and 21 CTRL samples in the space of the first two principal components. In Figure 1A a relatively good separation of the DLBCL lipidomic landscape from that of healthy females is visible, with 13 points residing in the well-separated DLBCL data space, 2 points being close to the CTRL data space and only 2 DLBCL samples misplaced. Furthermore, both sets of data, DLBCL and controls, were more or less uniformly distributed over the data space pointing out to the absence of a major subclustering. In other words, no subsets of DLBCL samples (besides two close points in the upper right quadrant) due to potential confounding factors that would affect the measured lipid levels, such as age, diet, lifestyle factors, were present in the cohorts. The equal spread of data in both groups also implies a homogeneity of variance, thus making the univariate data analysis more accurate. A total of seven principal components that cumulatively explained 68% of the total variance in the dataset were selected as directions of the data that captured a maximal amount of variance (Appendix A), as also shown in the scree plot in Appendix A.

Besides score plot, loadings of the principal components were computed. The loading shows a correlation between a variable and a particular principal component, goes from −1 to 1 (zero meaning no correlation), where the bigger the absolute value of the loading, the greater the influence of that variable on a component. The clustering of lipid variables in Figure 1B,C is obvious. The first principal component encompassed a total of 75 lipid features with loading values over 50%, and these lipids appeared as clusters: (i) 18 lipid species with bound LA: 10 PCs, 4 ether-linked PE species (PE plasmalogens), 2 LPEs and 2 PIs; (ii) 18 lipid species with bound AA: 10 PCs, 4 PE O-species, 3 PIs and one LPE species; (iii) 19 SM species + Cer d40:2; (iv) 4 PLs containing DHA (Figure 1B). This data clearly suggests the role of PL precursors of LA and AA, as well as major cell membrane SL species, in the pathobiochemistry of DLBCL. The second principal component contributing much less to the total variance in the dataset covered the two clusters of SLs: (i) seven SMs including three species confirmed to contain d18:1 sphingosine (SPH) such as SM d36:1 (d18:1_18:0); (ii) five Cer species (one is Cer d18:1_16:0), and one cluster of four PEs + PE plasmalogene species containing, again, LA or AA (Figure 1C). Cumulatively, data from the PCA loadings suggests the involvement of PLs containing PUFAs: LA, AA and DHA alongside SMs containing SPH d18:1 that could serve as precursors of major pro-inflammatory (20:4n-6), anti-inflammatory PUFA (22:6n-3) and S1P in the pathobiochemistry of DLBCL. In other words, PCA, although a pure mathematical model, sensed the metabolic interconnections between AA, LA and DHA. 

To further visualise and dissect differences in plasma lipidome between DLBCL and control females, an OPLS-DA model was built out of 155 lipid features and 38 observations (model details in Appendix A). The distinct group separation was completely achieved using an OPLS-DA approach (Figure 1D). It was a good fit for the lipid dataset with a goodness-of-fit R2 = 0.758. The predictive power of the model was Q2 = 0.513, a reasonably high value, and its statistical significance confirmed that the model was valid (Appendix A). 

An S-plot was drawn in order to visualise the plasma lipid species with the highest concentration differences between DLBCL and control samples (Figure 1E). The S-plot is often used to find putative biomarkers from an omics dataset. The further the lipid variable is away from the origin, the higher the VIP value. In concordance with data from the untargeted lipidomics presented in the S-plot, we suggest the importance of the following five plasma lipid species: PC 18:2_18:2, PC O-18:1_20:4, SM d44:2, Cer d44:1 and Cer d18:1_24:1 in discriminating DLBCL from healthy status in females. We see, again, the important role of LA- and AA-containing PC species, alongside two SPH d18:1-containing Cer species, in differentiating the two studied cohorts. It is a common practice in lipidomic studies to extract via OPLS-DA another parameter, known as variable of importance in projection (VIP), to rank the overall contribution of each lipid feature to the model. Variables with VIP > 1 or >1.5, depending on the number of measured lipid species, are considered relevant for the discrimination between the studied groups [33]. The plasma lipid species with the greatest VIP values were as follows (Appendix A): PC 18:2_18:2 (VIP 2.25), PI 18:1_18:2 (2.04), PE 18:0_18:1 (2.01), PC 18:0_18:2 (1.97), PC 18:2_20:4 (1.94), LPE 18:2 sn-2 (1.93), PE O-16:1_18:1 (1.90), PE O-16:1_18:2 (1.90), LPC 18:2 sn-1 (1.84), Cer d18:1_24:1 (1.82), Cer d44:1 (1.81), PI 18:0_18:1 (1.80), PS 18:0_18:1 (1.73), PE O-17:1_18:2 (1.66), PC 14:0_18:2 (1.63), PI 18:1_ 20:4 (1.62), PE O-20:1_18:2 (1.61), PC 16:1_18:2 (1.61), PC O-18:1_20:4 (1.60), LPE 18:2 sn-1 (1.56), SM d39:1 (1.53) and PC 18:1_18:2 (1.51). Similarly to PCA, in building a model, OPLS-DA takes a whole dataset into an account. This approach again emphasised the mechanistic role of PL and SL precursors of lipid mediators in DLBCL in women. 

It appears that differentially expressed features in a comprehensive omics experiment are more readily identified by using an FC than a t-statistic, similarly to microarray assays [63]. Given that lipids are produced and degraded in enzymatic reactions and genes coding for enzymes in lipid pathways are often differently expressed in cancer than in healthy cells and tissues [13,64], FC of lipids also appears as a more meaningful discriminator of cancer and healthy status than *t*-test. To assess the importance of lipid species with high or low FC values, besides low *p*-values, a Volcano plot was drawn (Figure 2). This time, more plasma lipid species were revealed as DLBCL specific: PC 18:2_18:2, PC 18:0_18:2, PI 18:1_18:2, PI 18:0_18:1, LPC 18:2 sn-1, LPE 18:2 sn-2, PE O-16:1_18:1, PE O-16:1_18:2, PE 18:0_18:1, PC 18:2_20:4, SM d39:1, SM d41:2, PS 18:0_18:1, Cer d44:1, Cer d18:1_24:1, Cer d43:2, LPE 22:5 and PE O-22:2_20:4. Finally, if we choose the most profoundly dysregulated lipid species in DLBCL based on the fulfilment of these three criteria, *p* < 0.05, FC > 20% for molar concentrations and VIP ≥ 1.5 [33], these would be the following 21 lipid species: PC 14:0_18:2, PC 16:1_18:2, PC 18:0_18:2, PC 18:1_18:2, PC 18:2_18:2, PC 18:2_20:4, LPC 18:2 sn-1, SM d39:1, Cer d18:1_24:1, Cer d44:1, PE 18:0_18:1, PE O-16:1_18:1, PE O-16:1_18:2, PE O-17:1_18:2, PE O-20:1_18:2, LPE 18:2 sn-1, LPE 18:2 sn-2, PI 18:0_18:1, PI 18:1_18:2, PI 18:1_20:4 and PS 18:0_18:1.

### 3.3. Inflamatory Lipidomic Profile in Females with DLBCL

The untargeted lipidomic profiling of DLBCL plasma revealed a general change in the glycerophospholipidome AA pool with possible consequences on downstream inflammatory cascades of the COX and lipoxygenase (LOX) pathways. Moreover, considering the significant changes in the abundance of many LA- and AA-containing PLs and known remodelling and signalling routes, it appears that the most commonly affected biochemical/cellular processes were mobilisation of LA and AA from the PC and PE species, emptying the plasma reservoirs of SLs (to produce SPH and possibly S1P), and interplay among PUFAs possibly linked to PI3K pathway. In order to elucidate how such downstream processes infer metabolic and signalling pathways involved in the etiology of DLBCL, we designed and conducted a targeted lipidomic study of 35 lipid species relevant to the PI3K signalling and eiCs-related inflammatory processes. 

#### 3.3.1. Overview of Changes in the Inflammatory Lipidome of Plasma in DLBCL

The vast majority of targeted plasma lipids, eiCs, PUFAs, SMs, S1P and almost all PLs (except LPCs) showed a tendency to be higher in the DLBCL cohort than in the CTRL, but the statistical tests failed to recognise significance due to a great within-group variance (Appendix A). The only exception was PI 34:1, with the mean value significantly lower in DLBCL than in CTRL (Figure 3).

Twelve lipids passed the threshold of statistical significance (Appendix A), but after the BH correction, only ten of them were considered: S1P, SM 36:1, SM 34:1, PI 34:1, LPI 20:4, 12(S)-HETE, 15(S)-HETE, 20-HETE, LA and PS 38:4. Hence, the most profoundly altered lipids in the plasma of DLBCL female patients were those from the SL metabolism (Figure 3 and Appendix A): S1P, SM 36:1 and SM 34:1 (https://www.genome.jp/pathway/map00600, accessed on 5 March 2023). Essentially, the ten lipids belong to the group of lipid mediators, i.e., they are directly involved in signalling (S1P, LPI 20:4 and HETEs). Others are metabolically linked to lipid mediators (SM 36:1 and SM 34:1 to S1P), or serve as precursors of lipid mediators such as LA. PI 34:1 is a precursor of second messengers in the PI3K signalling pathway.

DLBCL and CTRL females were not exactly age-matched, mainly because this cancer hits older persons and it is almost impossible to find older controls, i.e., females who do not have insulin resistance, metabolic syndrome, dyslipidemia or take some medicines. Plasma lipidome exhibits age-related changes stemming from metabolic perturbations, oxidative stress and chronic inflammation. Age-linked lipid changes include: S1P (−0.44% change per year), SM 34:1 (0.29%), SM 36:1 (0.22%), PI 34:1 (−0.19%), LPI 20:4 (−0.03% at sn-1, 0.26% at sn-2) and PS 38:4 (−0.12%) [65]. The analytes we also measured in our study and found as significantly altered in DLBCL, but the relevant increases outweighed the reported age-related changes. In the case of plasma levels of S1P and PS 38:4, we even managed to observe their increase in DLBCL (Figure 3 and Appendix A), although the age-related decrease was anticipated in the work of others [65]. Therefore, although the DLBCL patients had a greater mean age than the healthy controls in our study, we do not expect them to confound the results, as it is known that DLBCL patients, as a rule, have lower cholesterol and FAs than expected for their age [66,67].

The within-group variance for 15 lipids was significantly higher in the DLBCL sample group than in the CTRL: S1P, SM 34:1, 5(S)-HETE, 5(6)-DiHET, 15(S)-HETE, LPI 20:4, SM 36:1, PC 38:4, LA, 8(9)-DiHET, 11(12)-DiHET, PS 38:4, AA, SM 42:1 and PE 38:4, (Appendix A). Although the plasma level of TxB2 was 3.12-fold higher in the DLBCL than in the controls, the change was still non-significant. The plasma content of LA, EPA and DHA in DLBCL was approximately 1.5-fold higher than in CTRL, but only a change in LA was found to be significant (Appendix A). Therefore, the change in LA detected by the targeted lipidomics confirmed our hypothesis regarding the extensive mobilisation of this PUFA from its plasma reservoirs in the form of PLs, particularly PCs (obtained from our untargeted lipidomics). A concerted increase in omega-3 PUFAs could stem from their role in the compensatory anti-inflammatory mechanisms of the immune response to the cancer insult. 

#### 3.3.2. Modelling Plasma Lipid Mediators in Female Patients with DLBCL

We assessed a possible clustering of plasma samples according to the health state (DLBCL vs. healthy) through visualisation of principal component scores for the two groups, using the dataset obtained through the targeted lipidomics of the 35 lipid species, much as we did with the untargeted lipidomics data (see Figure 1). The plot combining the first two principal component scores for the targeted lipidomic analysis is displayed in Figure 4. 

Five principal components, PC1 to PC5, were selected that cumulatively explained 65.5% of the total variance (Appendix A).

Although the complete group separation was not evidenced through the score plots, the healthy group formed a, subspace within the wider space occupied by the DLBCL samples (Figure 4A). According to the PCA, it appeared that a clear-cut line between the plasma lipidome of controls and DLBCL females could not be drawn, which was also the case with the score plot in Figure 1A. In comparison to healthy females, the lipid concentrations in the DLBCL cohort were more distributed across the space of values. We rationalise that this was likely due to different origins of lipids, with different cell populations of tumour and TME being metabolically non-uniform, leading to the larger within-group variance of their (inflammatory) lipid levels. This also extended to their lipidomic signatures, rendering their PCA score plots indistinguishable from controls. 

Similarly to the loadings plot obtained through the PCA of the untargeted lipidomics dataset (Figure 1B,C), the clustering of lipid species belonging to same classes or subclasses in Figure 4 is obvious. The majority of eiCs had great loadings and formed a cluster under the first principal component PC1 (Figure 4B), including AA and several eiCs: 5(S)-, 8(S)-, 15(S)- and 20-HETE, with all four DiHETs. EiCs originate from the common metabolic pathway of AA (Appendix A). However, TxB2 and 12(S)-HETE did not cluster with the majority of eiCs under the first principal component (Figure 4B). The second cluster of lipids comprised PC 38:4 (PC 18:0_20:4), PC 40:4 (PC 20:0_20:4), PI 36:4 (PI 16:0_20:4) and PI 38:4 (PI 18:0_20:4) (Figure 4C). The common features among these PLs are that they belong to PC or PI lipid subclass and contain a saturated FA and AA, the latter most likely sn-2 bound. On the other hand, all PUFAs: LA, EPA and DHA clearly grouped also under the first principal component (Figure 4B), which could be due to the metabolic interconnections between AA, LA and n-3 PUFAs: EPA and DHA. The interplay among the pro- and anti-inflammatory n-6 and n-3 PUFAs and their downstream metabolites may contribute to inflammatory conditions in DLBCL. The principal component PC2 explained approximately 15% of the total variance and revealed two metabolically associated lipid clusters (Figure 4C): (i) PL species that did not group under the first principal component, such as PG 34:1 (PG 16:0_18:1), PE 34:1 (PE 16:0_18:1), PE 38:4 (PE 18:0_20:4), PE 38:6 (PE 18:2_20:4) and PS 38:4 (PS 18:0_20:4) and SLs, that is three sphingomyelins: SM 34:1, SM 36:1, SM 42:1 and S1P. The former share some common features: oleic acid (FA 18:1) or AA, most probably at sn-2 and three out of four possess an ethanolamine polar head (Figure 4C). 

PCA revealed a general differential plasma lipid pattern of DLBCL, which is consistent with a generalised disruption of lipid homeostasis and signalling in this cancer. PCA took into account the lipid dataset as a whole, thus overwriting the mere statistical differences in individual lipid levels between DLBCL and CTRL females. Noteworthy, the unsupervised algorithm, PCA sensed metabolic links among different lipid pathways, like those that connect SL and PL pathways to the AA/eiCs pathway, S1P signalling routes and the PI3K pathway. In other words, a mathematical model apparently revealed, to a certain extent, the metabolic logic behind the different plasma lipidomic profile in DLBCL vs. healthy state. 

As we did with our dataset obtained from the untargeted lipidomics, we subjected the 38 × 35 dataset (38 plasma samples, 35 lipid features) to an OPLS-DA algorithm to extract ten variables with the greatest VIP values (Table 3), and these were: S1P, SM 36:1, SM 34:1, PI 34:1 (VIP > 1.5), SM 42:1, PS 38:4, LPI 20:4, 12(S)-HETE, LA and RvD1 (1 < VIP < 1.5). This approach again emphasised the mechanistic role of major SM species and bioactive lipids/lipid mediators: S1P, 12(S)-HETE, LPI 20:4, RvD1 and AA-containing PS 38:4 (can be obtained by decarboxylation of PE 38:4) in DLBCL in women. 

Although the focus of our study was to explore the plasma lipidomic landscape of DLBCL in a pilot untargeted lipidomics study, as well as a critical set of lipid mediators and signalling lipids involved in inflammation and immune regulation in the targeted study, the latter combined with the statistical/MDA also revealed lipid signatures of DLBCL (Table 3), which could be further explored as candidate biomarkers of DLBCL. Moreover, the combined untargeted and targeted lipidomics helped us to infer mechanistic aspects and pathways affected in this cancer. 

### 3.4. Correlation Analysis of the Lipid Mediators and Precursors

Spearman correlation coefficients that were found to be significant in DLBCL and CTRL groups are shown in Appendix A. 

In the DLBCL group, 14 correlations between measured lipid species were significant, which was much less than the number of significant correlations in the CTRL group (29 correlations). There were some new strong correlations between plasma lipid species in the DLBCL group, such as AA vs. 8(S)-HETE, AA vs. EPA, PC 40:6 vs. PE 38:6, LA vs. DHA, PC 38:4 vs. PC 40:6 and 11(12)-DiHET vs. DHA, which were not statistically significant in the CTRL. Another eight significant correlations in the DLBCL group also appeared as significant in the CTRL group. On the other hand, there were 29 significant pairs of lipid species in the healthy group and some of the most pronounced correlations were lost in the DLBCL group, for example: PC 38:6 vs. DHA and PC 40:6 vs. DHA (Appendix A). It can be seen that some lipid species in the DLBCL group were engaged in more than one significant correlation: AA (three), DHA (three), PC 40:6 (three), 5(S)-HETE (two), 8(S)-HETE (two), EPA (two), 11(12)-DiHET (two) and SM 36:1 (two). In addition, the majority of significantly correlated lipid species in DLBCL group were PUFAs (eight correlations), eiCs (six correlations), or PLs containing AA (PC 38:4, PC 38:6, PC 40:6, PE 38:6 and PI 36:4). PC and PE are the most abundant PL species in cell membranes and, hence, the most readily available source of AA in the case of activation of immune cells. 

Heatmaps were created that depict differential relationships among measured plasma lipid species in DLBCL cohort and CTRL females (Appendix A).The correlation network showing the most dysregulated plasma lipids in DLBCL female patients in comparison with the CTRL group is depicted in Figure 5. Not only was S1P identified here as the most altered lipid species in the DLBCL cohort, but its significant negative correlations with AA and LA, which existed in the control group, disappeared in the DLBCL cohort (Figure 5). Similarly, 12(S)-HETE was in significant correlation with two AA-containing PI species: PI 36:4 and PI 38:4 only in plasma of CTRL women (Figure 5A). LA was linked to AA and 5(S)-HETE (entry for LA in the AA pathway is shown in Appendix A), and also to S1P (a negative correlation in Figure 5A). In DLBCL, though, LA was only related to DHA (Figure 5B). TxB2 showed no significant correlations whatsoever, much like RvD1 and 20-HETE. 

Finally, Table 3 depicts all significantly changed plasma lipid species in DLBCL female patients in comparison to healthy control females.

## 4. Discussion

DLBCL is a fast-growing B-cell lymphoma that is potentially curable if diagnosed early, but the prognosis also depends on the molecular subtype of the tumour [8]. Numerous alterations in the genetic landscape of DLBCL have been reported that widely reflect on a spectrum of signalling cascades causing their dysregulation [68,69]. The most important cellular processes that are disturbed in DLBCL are those involved in B-cell receptor (BCR) signalling and differentiation, activation of the oncogene cascades such as NF-κB and PI3K/AKT/mTOR pathways, apoptosis and epigenetic regulation. In order to expand our understanding of DLBCL mechanisms, a lipidomic landscape would complement with new valuable molecular insights, the findings and clues from genetics/epigenetics. Lipids emerge as essential molecules involved in each and every step of lymphoma genesis, maintenance and metastasis.

In the pilot study presented herein, we explored the lipidomic landscape of plasma from 17 therapy-naïve female patients with DLBCL (Table 1) and 21 disease-free females, in an untargeted lipidomic analysis covering 155 PL and SL species, including PCs, LPCs, PEs, LPEs, PIs, LPIs, PSs, plasmalogens, SMs, and Cer. In addition, in a targeted lipidomic study of the same cohorts, we measured levels of major lipid mediators revolving around the AA and its downstream metabolites and precursors: eiCs, RvD1, LPC 18:0, LPC 20:4, LPI 20:4, two major SMs (SM d18:1_16:0 and SM d18:1_18:0), S1P, four major PUFAs: AA, LA, EPA and DHA, as well as several AA-containing PLs: PC, PE, PI, PS and PG.

In our untargeted lipidomic study, we detected a major depletion of both the PL and SL plasma pool in DLBCL, except for a few plasmalogen and Cer species, respectively. The levels of 54 PL and SL species (Table 2) were found to decline with DLBCL, which supports the essential role of metabolic lipid perturbations in DLBCL. The profound down-regulation of plasma PCs and SLs relative to controls in patients with AML, alongside the decline in triglycerides and cholesterol esters, was proposed to be driven by an enhanced rate of FA oxidation (FAO) in AML [38]. Although we did not profile triglycerides and cholesterol esters by lipidomics, their total values measured in DLBCL plasma were in the normal range, so we cannot suggest an augmented rate of FAO in our DLBCL cohort. 

Upon a close look at the affected species in DLBCL plasma (Table 2), we notice that a great number of the down-regulated PL species carried either 18:2(n-6), that is LA, or 20:4(n-6), that is AA. There is a link between the extent of AA accumulation in cell membrane PLs and the ability of inflammatory cells to produce enough quantities of pro-inflammatory eiCs [70]. Knowing this, we assumed that the activation of the AA pathway in DLBCL with a massive production of lipid mediators, presumably eiCs, accompanied the massive drop in plasma PLs. If our hypothesis is true, we should be able to detect the increased levels of AA and LA in DLBCL plasma. In the targeted approach, we actually did detect a significant rise in LA in DLBCL plasma, but no change in the AA level was detected. The latter may be due to the higher within group variance in the DLBCL than in the control group. However, we have to take into account that the level of AA, being a central figure of the potent pro-cancer and pro-inflammatory signalling cycle, must be maintained in all circumstances. There are three main mechanisms through which cancer cells can obtain AA: (1) direct exogenous uptake, (2) de novo synthesis from LA and (3) cleavage of the sn-2 positions of those membrane PLs that carry AA [71]. Though we found a significantly decreased levels of AA-containing PLs in the DLBCL plasma by the untargeted approach, the decline was even more pronounced for the LA-containing PL species, particularly the PCs (Table 2), with AA-containing species coming second. That is why we here suggest the augmented rate of synthesis of AA from LA in DLBCL, by the action of relevant elongases and desaturases [71]. The increased rate of de novo synthesis of AA from LA was also suggested to be a source of AA in AML plasma [38].

It is the task of phospholipases A1 and A2 to detach the esterified LA and AA from the sn-1 and sn-2 positions of PLs. The individual secreted sPLA2s have distinct substrate selectivity for the polar head groups or sn-2 bound FAs of their substrate PLs [72], and a role of sPLA_2_ in driving lipid metabolism towards pro-tumourigenic actions in B-cell lymphoma has been recently reported [73]. Therefore, provided that there is a more or less continuous supply of AA, biosynthesis of eiCs in DLBCL cancer cells can render a chronic low- to high-grade inflammation. For this to happen the following three groups of enzymes have to be properly expressed and active: LOXs to produce HETEs, COX-2 to produce prostaglandins (PGs) with TxB2 at the end of one PG biosynthetic road and cytochrome P450 ω-hydrolases (CYPs) to produce 20-HETE and DiHETs (Appendix A). 

COX-2 is frequently expressed in many types of cancers, playing a multifaceted role in the promotion of carcinogenesis and cancer cell resistance to chemo- and radiotherapy. This enzyme is not only produced by cancer cells but also by cancer-associated fibroblasts and specific macrophages in the TME [74]. It seems that all the three pathways functioned well in our DLBCL cohort, since we observed a tendency for almost every eiC in DLBCL plasma to be increased, except for 11(12)-DiHETs (Appendix A). However, these tendencies were statistically significant only for 12(S)-HETE, 15(S)-HETE and 20-HETE (Table 3). The most prominent alterations were those of 12(S)-HETE (3.15-fold) (Table 3), probably reflecting an augmented rate of arachidonate 12-LOX (12-LOX) activity. Considering the overexpression of 12-LOX and its product 12(S)-HETE in one haematological tumour, chronic myeloid leukaemia [75], the prominent rise in plasma content of 12(S)-HETE seen in DLBCL female patients (Table 3) did not surprise us. Enzyme 12-LOX (also known as platelet-type ALOX 12) is coded by *ALOX12* located on the chromosome 17p, along with all other genes encoding LOXs, i.e., *ALOX15B*, *ALOX15, ALOX12B* and *ALOXE3*, the only exception being 5-LOX. Human 12-LOX and 15-LOX 1 are 12/15-LOXs because they are able to convert AA to both 12(S)-HpETE and 15(S)-HpETE, which are precursors of 12(S)-HETE and 15(S)-HETE, respectively (Appendix A), and they conduct this same metabolic conversion on free AA, and also on AA esterified to membrane PLs. On the other hand, human 15-LOX 1 enzyme can generate both 12(S)- and 15(S)-HpETE, whereas 15-LOX 2, encoded by *ALOX15B* is not able to make 12(S)-HpETE, but only 15(S)-HpETE [76]. 15-LOX 2 and 12-LOX were reported to exert tumour suppressor roles in murine lymphoma [76,77]. It is tempting to speculate that the increased levels of 12(S)-HETE and 15(S)-HETE observed in our study served to cope with the tumour. 

The role of the AA pathway and inflammation in the pathogenesis and progression of B-cell lymphomas including DLBCL has been increasingly recognised [78]. Interestingly, while tumour cells derived from 85% of Hodgkin lymphomas expressed 15-LOX 1, the enzyme was not detected in any of the ten biopsies that represented nine different subtypes of NHL, which makes 15-LOX 1 a candidate biomarker to distinguish Hodgkin from NHL [79]. The precise role of 15(S)-HETE is still not elucidated, partly because 15-LOXs are also involved in the synthesis of D-series resolvins from DHA [39,80]. One of them, RvD1, is a potent anti-inflammatory lipid mediator [39]. In our study, plasma levels of 15(S)-HETE and RvD1 were approximately 1.4-fold higher in DLBCL than in controls, and statistically significant (Table 3 and Appendix A). It is probably not a coincidence that both of these lipid mediators were made by 15-LOXs. RvD1 rise in DLBCL plasma suggests a compensatory homeostatic mechanism in response to the pro-inflammation. RvD1 is synthetised from DHA (22:6(n-3)), and we detected a significant down-regulation of few PL species carrying this FA by the untargeted lipidomics (Table 2). 

A very recent work details how deficiency in *ALOX15B* (due to deletions in 17p chromosome) contributes to B-cell malignancy [81]. The authors found all metabolites of the LOX pathway to be reduced due to the 15-LOX 2 loss [81]. We found elevated plasma levels of 15(S)-HETE in DLBCL (Table 3), which meant that 15-LOX 2 was active and genetic deletions in the mentioned region on 17p chromosomes did not occur. The eiC levels in our study and those of Qi et al., 2023, are not expected to overlap, since we measured lipid species in plasma of DLBCL patients and they used lymphoma cell lines and leukocytes isolated from the patients with B-cell chronic lymphocyte leukaemia (B-CLL) [81]. In addition, the lymphomas were different in the two studies, DLBCL vs. B-CLL. In the study of Qi et al., 2023, AA levels in the B-CLL cells were up-regulated, along with AA metabolites of the COX pathway, mostly PGE2 and TxB2, due to the compensatory activation of COX-2 [81]. In our study, though, the level of AA in DLBCL plasma was not significantly altered compared to controls (Appendix A). Although we cannot be sure of the enhanced activity of COX-2 in our DLBCL cohort, we detected a substantial rise in plasma TxB2 levels (3.12-fold), but it was not statistically significant due to the great within-group variance (Appendix A) and the massive decline in the plasma pool of AA-containing PLs supports this hypothesis (Table 2). We infer higher rates of 12-LOX, 15-LOX 2 and CYP 4A or CYP 4F activities from the increased levels of 12(S)-HETE, 15(S)-HETE and 20-HETE in the DLBCL plasma that are responsible for the consumption of plasma reservoirs of PLs. We hypothesise that the profound down-regulation of plasma PLs stems largely from the activation of COX-2. The expression of COX-2, the enzyme responsible for converting AA into pro-inflammatory eiCs, is often up-regulated in patients with NHL [82]. COX-2 is enriched in lymphoma cancer cells, and high expression of *PTGS2* that encodes COX-2 coincides with a dismal outcome in lymphomas [82]. Furthermore, it seems that a cross-talk between COX-2 and 15-LOX 2 is involved in the pathobiochemistry of lymphoma [81].

Eberlin and colleagues (2014) observed distinct lipid signatures in human lymphoma tissue samples, depending on the extent of MYC oncoprotein expression. In their study, 86 PL species were found to be up- or down-regulated. For example, 40 PLs were found to be increased in MYC-induced lymphomas, but PE, PS and PI species were generally decreased [83]. All DLBCL tumour samples had a high expression of MYC oncogene, which is a global regulator of transcription in cancer, including cellular growth and lipid biosynthesis. MYC is especially involved in the pathogenesis of haematopoietic tumours and lymphomas and is also responsible for the FA import for FAO, the latter being an important process for lymphoma cell survival. MYC gene alterations have often been detected in mature B-cell neoplasms and are usually associated with an aggressive clinical behaviour. Since we excluded the high-grade lymphomas from our DLBCL cohort, no involvement of MYC should be expected, hence we did not expect to see a great extent of similarity between the (phospho)lipidomic landscape in DLBCL plasma in our study (Table 2) and that obtained by Eberlin and collaborators [83]. To sum up, the importance of the AA cascade in the progression of DLBCL is evidenced by both our untargeted and targeted lipidomics data.

In order to enable increased plasma levels of LA and AA needed to sustain a pro-inflammatory state in DLBCL, in our cohort, tumour cells were most likely in a state of increased fatty acid synthase (FASN) activity. This multienzyme complex is responsible for de novo biosynthesis of FAs, with palmitate (16:0) at the end of this metabolic route. Augmentation of de novo FAs synthesis, first observed in solid cancers, occurs also in haematological malignancies [84]. The expression of FASN was found to be enhanced in most DLBCL tumour samples and also in cell lines of DLBCL [85,86]. FASN overexpression is associated with the aggressive clinical course of DLBCL [87]. 

Interestingly and in line with our findings, FASN levels seem to be related to the activation of the PI3-kinase (PI3K)/AKT pathway. In fact, FASN-induced PI3K-S6Kinase signalling promotes the oncogenic translation in DLBCL [88]. Particularly, GCB-like DLBCL cells (FASN-resistant) are dependent on PI3K signalling [88]. Point mutations of a catalytic subunit of enzyme PI3K, p110δ, were described in a panel of DLBCLs [89]. Hence, in the situation where the PI3K is overexpressed or p110δ overactive, PI reserves in the plasma membrane of immune cells might be converted to PIP_2_ and end up as PIP_3_. Using the targeted lipidomics, we found significant alterations in the levels of both PI 34:1 (PI 16:0_18:1) and LPI 20:4 in DLBCL plasma in comparison to the controls (Figure 3 and Table 3). The abundance of LPI 20:4 was 1.45-fold greater in the DLBCL plasma, while the PI 34:1 level was 1.6-fold lower (Table 3). The lower level of the latter was most likely due to the metabolic conversion of this major PI lipid (PI 16:0_18:1) into PIP_2_. The sharp drop in PI 34:1 (16:0_18:1) detected by the targeted lipidomics (Table 3, FC 0.62) was also observed by the untargeted lipidomics (FC 0.63, Table 2). Accordingly, based on the literature data mentioned above and decreased plasma levels of PI 34:1 observed in the DLBCL cohort by us (Figure 3, Table 3), we can speculate that this cancer used the plasma reservoir of PI 34:1 to produce second messengers of the PI3K pathway. The significant decline in PI levels in DLBCL plasma also occurred with few other PI species (Table 2), thus confirming our hypothesis.

LPI 20:4 sn-2 is formed mainly from PI 38:4 (PI 18:0_20:4) through the action of the enzyme PLA1. We expect that much smaller quantities of plasma LPI 20:4 originated from PI 36:4 (PI 16:0_20:4), because the latter is 6- to 8-fold less abundant than PI 38:4 (Appendix A) and [90]. We noticed that the rise in plasma LPI 20:4 was not accompanied by the decline of PI 38:4 or PI 36:4 levels (Appendix A). These results are confirmed by the untargeted approach (Appendix A). Given that in DLBCL the mean plasma level of PI 38:4 is 15 times that of LPI 20:4 (Appendix A), the latter could have been produced from PI 38:4 under the radar of the statistical significance. There was a slight tendency for PI 38:4 levels to be smaller in the DLBCL group (Appendix A). Possibly, the observed rise in LPI 20:4 originated from PI 38:4 (and PI 36:4) and/or some other plasma PI species, which could have been quickly replenished through the remodelling (diacylation/reacylation) of PI to ensure a continuous supply of LPI 20:4. 

A high expression of a G-protein-coupled receptor GPR55, which is a cognate receptor for LPI, appears to correlate with cancer aggressiveness in AML, another haematological cancer [91]. We suggest, based on the reported functional role of LPI 20:4 in immune response and cancer progression [92], that the reason cells that circulate in the plasma of DLBCL (tumour + TME cells) produce more LPI 20:4 is to sustain cancer cell growth and evade the host immune response. A recent research established a link between activated S1P receptor and LPI receptor GPR55, which interacted strongly and specifically, both being erroneously expressed in many cancer types [93]. Hence, the augmented plasma levels of both S1P and LPI 20:4 in DLBCL detected in our study (Table 3) point towards a concerted modulation of the GPR55 by S1P and LPI. 

S1P is a bioactive sphingolipid metabolite, a promotor of cell growth and survival, and is evidenced to play the role of pro-cancer signalling molecule in haematological cancers [94]. This signalling lyso-sphingolipid and other SLs are metabolically interrelated: S1P is generated from SPH, through the action of two inducible enzymes responsible for S1P synthesis, sphingosine kinases SPHK1 and SPHK2. Enzymatic hydrolysis of membrane SLs by sphingomyelinases (SMase) gives rise to Cer, and Cer may be converted into SPH by the action of ceramidases [47].

The intracellularly generated S1P from cell membrane SM precursor is secreted out of the cancer cell and extracellular S1P becomes a ligand for GPRs S1P_1-5_ [47]. S1P concentrations in cancer tissue are almost always higher than those in surrounding non-cancerous tissue [95,96,97]. Importantly, SPHK1 is over-expressed in DLBCL [50]. Based on these reports, the observed increased abundance of S1P in the plasma of DLBCL female patients (Table 3) was expected. It is worth noting that the mean SPHK1 expression in NHL tumour tissue was found to be enhanced by 1.45-fold compared to the control tissue [98], whereas the mean S1P in DLBCL plasma measured by us was 1.85-fold greater than that in the control plasma (Table 3), which represents an excellent agreement with the results of Bayerl et al. [98]. This would also imply that the increment in plasma S1P level detected in our study originates from the DLBCL and/or TME. 

The rise in plasma S1P levels in DLBCL female patients compared to control females was accompanied by the rise in two major SM species that carry the most abundant sphingoid base, SPH d18:1 along with the two most abundant saturated FAs, palmitic and stearic acid [90]. Hence, they were expected to serve as the major sources of free S1P. Additionally, the calculated ratios of SM/S1P were close to identical in DLBCL plasma and CTRL plasma (FC = 0.99, Appendix A), which led us to suggest that SM sourcing of S1P is maintained as increased in DLBCL to allow the increased production of S1P. In contrast to the substantial rise in SM 36:1 (d18:1_16:0) and SM 34:1 (d18:1_16:0) that we measured in the DLBCL plasma in our targeted approach (Table 3), we observed a sharp decline in fourteen SM species (Table 2) by the untargeted lipidomics, but none of these were SM 34:1 or SM 36:1. The three Cer species were up-regulated in the DLBCL plasma relative to the control cohort (Table 2). The rise in plasma Cer species along with the drop in SM species, as shown by the untargeted lipidomics (Table 2), may reflect the ceramide/SM homeostasis or SM cycle that takes place in the membranes of DLBCL cancer cells and other immunocompetent cells in its TME. 

The situation in a cancer tissue or cell may be opposite to what we observed in the tumour plasma if we assume that a tumour cell uptakes some lipids from the circulation and discards others. For example, the SM cycle in chemoresistant patient-derived leukaemia cells resulted in the maintenance of low Cer vs. high SM levels [99]. In the most recent lipidomic study on haematological malignancies, including B-cell lymphoma, myelodysplastic syndrome, AML and acute lymphatic leukaemia/lymphoblastic lymphoma, higher proportions of SM species containing FA 16:0 and FA 18:0 were reported [100], which is in accordance with our results (Table 3). However, the same group could not detect any change in the total serum SM concentration in these tumours [100]. In contrast, we found a significantly increased total plasma SM level in female patients with DLBCL compared to control females (*p*-value = 0.00047, Appendix A), probably because of different origins of B-cell cancers in their study and DLBCL in our present study. 

For the pilot study presented herein, we recruited 17 treatment-naïve female DLBCL patients, without comorbidities and without high-grade lymphoma. The rationale behind our choice of a single-gender lipidomic study was that we expected the gender differences to reflect on the plasma lipidome [101] and because of the limited availability of male participants. In this way, we avoided dividing the DLBCL cohort into dissimilar groups, which would lower the power of the statistical tests. Our study reveals that the metabolism of PUFAs, including the AA pathway, in females with DLBCL is affected by this cancer, hence a further investigation of nutrition influence on the cancer progression and comorbid inflammation is paramount. Another aspect of our study is that we have opened the door for future sphingolipidomic studies of this cancer. Further work will be required to dissect discrete mechanisms involved in these phenomena, but this falls beyond the scope of our current study. 

This combined approach, using untargeted and targeted lipidomics, while focusing on bioactive lipids that are involved in inflammation and immune response and their cell membrane precursors revealed the mechanistic aspects of the processes in DLBCL mediated by lipids. Moreover, we have also identified the major plasma lipid species discriminating between DLBCL and control females in both untargeted and targeted approaches. To the best of our knowledge, this pilot report is the first lipidomic study of plasma in DLBCL.

In summary, this combinatorial lipidomic investigation has uncovered some previously unrecognised metabolic and signalling pathways that were up- or down-regulated in DLBCL, and confirmed alterations in the lipid metabolic routes that were assumed or anticipated from genetic studies carried out mostly on mouse models of lymphoma. These findings can inform and help prioritise the prospective biomarker discovery that can expedite early diagnosis/prognosis of the disease. Moreover, we evidenced lipid signatures that can pinpoint the metabolic or signalling pathways predominantly affected by the disease hence opening novel perspectives on DLBCL management.

## 5. Conclusions

In the study presented herein, we employed a combinatorial untargeted and targeted lipidomics approach to identify major plasma lipid species discriminating between female DLBCL patients and disease-free control females, and to better understand the changes in lipid metabolism and the inflammatory context in this cancer. Because of a massive drop in the plasma content of LA- and AA-containing PLs, we suggest the activation of the AA pathway with an intensive production of eiCs. Furthermore, a substantial drop in the abundance of SLs in DLBCL plasma is concordant with the enhanced production of S1P, which is an important player in anti-cancer immunity. Besides an intensified de novo synthesis of AA from LA, we hypothesise that the DLBCL used the plasma reservoir of PI 34:1 to produce second messengers of the PI3K pathway. The results of our study on DLBCL strongly suggest a reprogramming of specific metabolic routes, including the altered turnover of PUFAs and increased rate of the production of lipid mediator LPI 20:4. To our knowledge, this is the first study that explores the plasma lipidome of DLBCL female patients. On the other hand, it is also the first study that combines profiling of carefully selected lipid mediators of inflammation and immunity with a broader lipidome profiling using the newest high resolution ion mobility mass spectrometry. Eicosanoids, PUFAs, sphingo- and phospholipids with signalling functions, i.e., S1P, LPI, Cer are of notoriously low abundance in plasma, usually requiring large volumes of plasma or enrichment, and many of these species are not detectable at all in the most modern high resolution unbiased lipidomic analysis. Only by using the combination of untargeted and targeted approach are we able to move the needle from mere lipid change alteration to a sound determination of pathways that are affected in this cancer, and hence help inform and focus future studies on new hypotheses and directions for cancer management. Lastly, this study also shows the power of lipidomics to unravel the essential mechanistic aspects of cancer biology. 

## Figures and Tables

**Figure 1 cancers-15-03653-f001:**
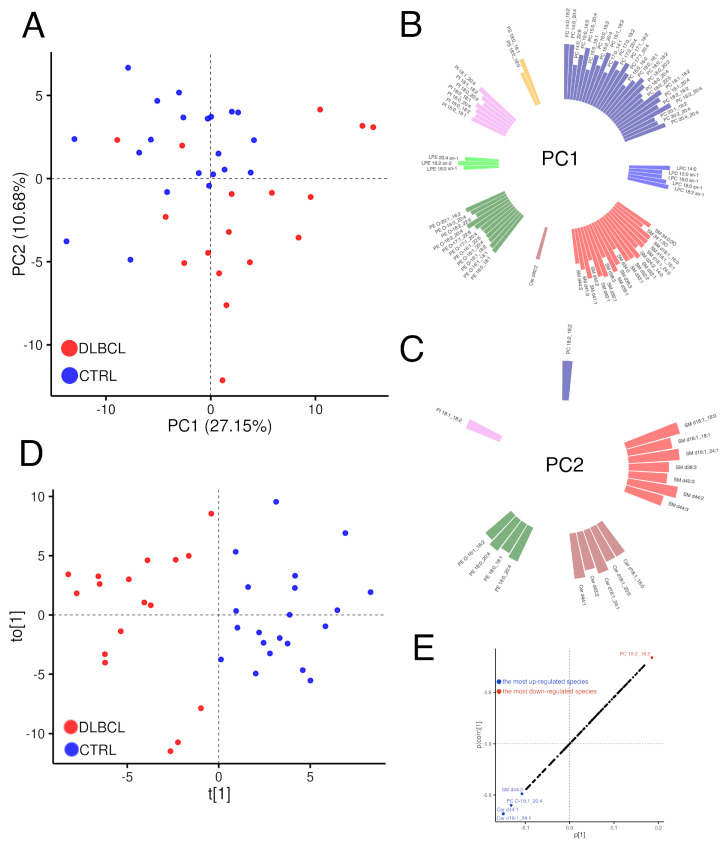
MDA for the untargeted lipidomics in diffuse large B cell lymphoma (DLBCL) and control (CTRL) plasmas. (**A**) Unsupervised PCA score plot to distinguish DLBCL from CTRL. Red and blue dots represent DLBCL and CTRL plasma samples, respectively (17 DLBCL + 21 CTRL). The proportion of total variance in the dataset explained by each principal component (PC1 and 2) is given in the brackets. (**B**,**C**) Loadings of the two major principal components: PC1 (**B**) and PC2 (**C**). The lipid variables that are highly correlated with principal components (>50%) are shown as coloured columns. (**D**) Supervised OPLS-DA score plot for the two components, predictive: t[1] and orthogonal in X: t_0_[1]. The models were built out of 155 lipid features semi-quantified in the targeted lipidomics (**E**). S-plot derived from the OPLS-DA model indicating the most down-regulated and up-regulated lipid species. p[1] axis describes the magnitude of each variable in X, while p(corr)[1] axis represents the reliability of each variable in X.

**Figure 2 cancers-15-03653-f002:**
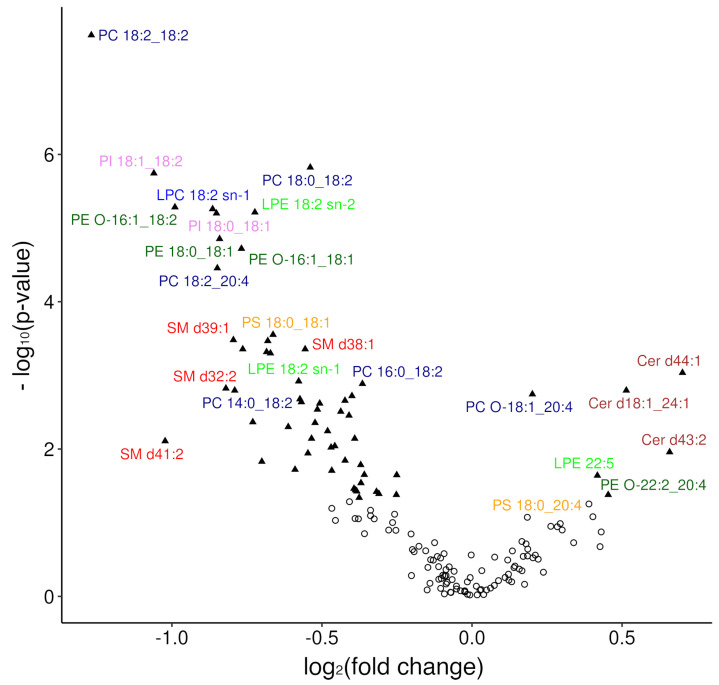
Volcano plot of −log_10_ (*p*-value) against log_2_ (FC). *p*-values related to the different abundance of a lipid species in DLBCL and control plasma were assessed by the univariate statistics of the untargeted lipidomics data (Table 2). FC is the ratio of mean values for the differential expressions between the two cohorts. Each point represents a lipid species measured, with triangles representing significantly altered lipid species (*p*-value < 0.05) and circles all others.

**Figure 3 cancers-15-03653-f003:**
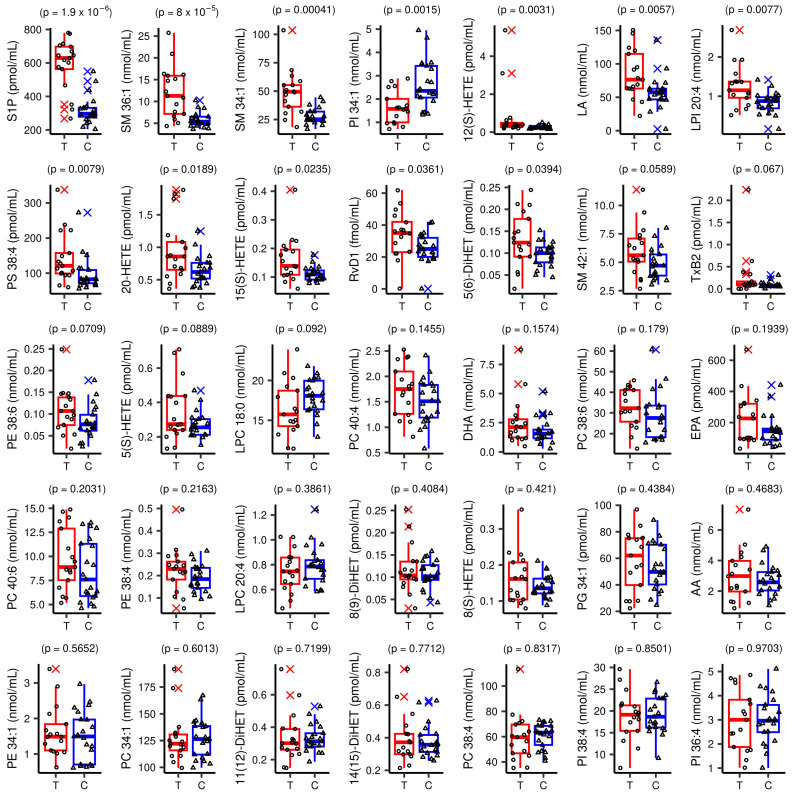
Circulating levels of lipid species in diffuse large B-cell lymphoma (DLBCL) patients (T, circles) and control subjects (C, triangles). In each box plot, the horizontal line represents the median, the box represents 1st and 3rd quartiles, and the whiskers are distances between the box and a value that stands at a maximal distance of 1.5*IQR from the box, and each point represents one observation/measurement. *p*-values obtained from between-groups statistical comparisons are also shown.

**Figure 4 cancers-15-03653-f004:**
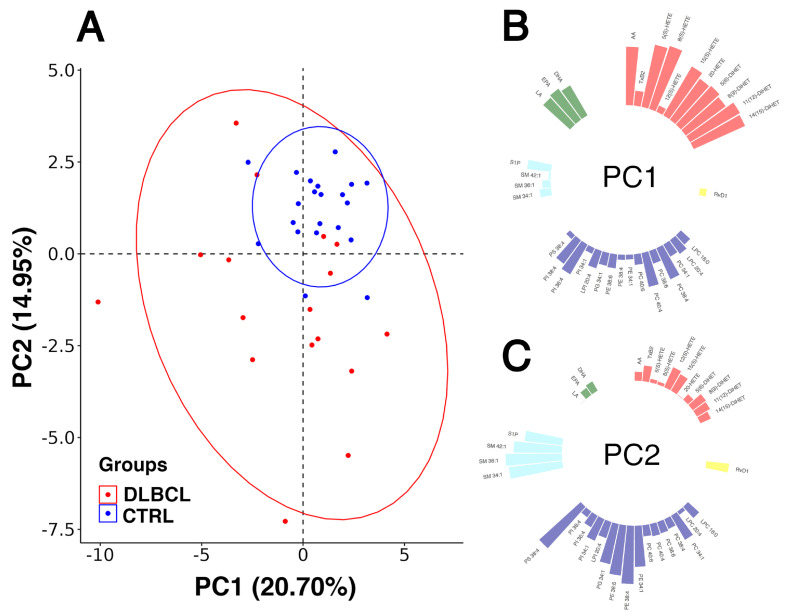
MDA for the targeted lipidomics in diffuse large B cell lymphoma (DLBCL) and control (CTRL) plasmas. (**A**) Unsupervised PCA score plot to distinguish DLBCL from CTRL. PCA was performed on the dataset obtained from the targeted lipidomics. Red and blue dots represent DLBCL and CTRL plasma samples, respectively (17 DLBCL + 21 CTRL). The proportion of total variance in the dataset explained by each principal component (PC1 and PC2) is given in the brackets. (**B**,**C**) Loadings of the two major principal components, PC1 (**B**) and PC2 (**C**). The lipid variables are shown as coloured columns.

**Figure 5 cancers-15-03653-f005:**
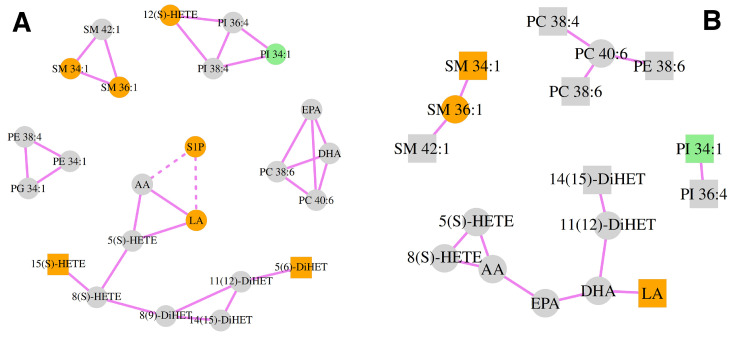
Correlation network of plasma lipid species in CTRL (**A**) and female patients with DLBCL (**B**). The line lengths relate to the extent of the correlation (ρ > 0.7), where straight lines indicate positive and dashed lines negative correlations. Circles represent the species that were in correlations with more than one other species. The most profoundly altered lipid species are coloured, orange denotes up-regulation, whereas PI 34:1 (green) was the only lipid less abundant in DLBCL than in CTRL.

**Table 1 cancers-15-03653-t001:** Clinical data on diffuse large B-cell lymphoma (DLBCL) female patients and healthy females (CTRL). AST, aspartate transaminase; ALT, alanine transaminase; GGT, gamma-glutamyl transferase; ALP, alkaline phosphatase; LDH, lactate dehidrogenase. The values are expressed as median (range).

Characteristics	Levels		Normal Values
	DLBCL	CTRL	
Age	59 (36–69)	49 (30–64)	
BMI	23.8 (19.4–32.2)	22.6 (19.6–29.5)	19.0–29.0
Triglycerides (mmol/L)	1.3 (0.6–2.6)		<1.7
Total cholesterol (mmol/L)	4.6 (1.4–6.8)		3.9–5.2
AST (IU/L)	17 (11–28)		<40
ALT (IU/L)	17 (6–36)		<41
GGT (IU/L)	17 (10–43)		8–61
ALP (U/L)	70 (39–116)		40–130
LDH	337 (246–633)		<250

**Table 2 cancers-15-03653-t002:** Comparisons of plasma lipid species between diffuse large B-cell lymphoma (DLBCL) female patients and healthy control females (CTRL). In total, 155 glycerophosphoholipid (PL) and sphingolipid (SL) species were measured in DLBCL cohort (n = 17) and CTRL (n = 21). Two-tailed *p*-values are shown: **** *p* < 0.0001, *** *p* = 0.0001 to 0.001, ** *p* = 0.001 to 0.01, * *p* = 0.01–0.05. Fold change (FC) for each lipid was calculated by dividing the mean concentration in the DLBCL group by the mean concentration in the CTRL group. Mean lipid levels in DLBCL plasma are either increased (↑) or decreased (↓) compared to CTRL.

Lipid Species	*p*-Value	Sig.	FC	Trend
PC 18:2_18:2	2.4 × 10^−8^	****	0.42	↓
PC 18:0_18:2	1.5 × 10^−6^	****	0.69	↓
PI 18:1_18:2	1.8 × 10^−6^	****	0.48	↓
PE O-16:1_18:2	5.2 × 10^−6^	****	0.5	↓
LPC 18:2 sn-1	5.5 × 10^−6^	****	0.55	↓
LPE 18:2 sn-2	6.1 × 10^−6^	****	0.61	↓
PI 18:0_18:1	6.3 × 10^−6^	****	0.55	↓
PE 18:0_18:1	1.4 ×10^−5^	****	0.56	↓
PE O-16:1_18:1	1.9 × 10^−5^	****	0.59	↓
PC 18:2_20:4	3.5 × 10^−5^	****	0.56	↓
PS 18:0_18:1	0.00028	***	0.63	↓
SM d39:1	0.00033	***	0.58	↓
SM d32:1	0.00034	***	0.62	↓
SM d38:1	0.00044	***	0.68	↓
PE O-17:1_18:2	0.00044	***	0.59	↓
Cer d40:2	0.00048	***	0.62	↓
PI 16:0_18:1	0.00051	***	0.63	↓
Cer d44:1	0.00092	***	1.63	↑
LPE 18:2 sn-1	0.0012	**	0.67	↓
PC 16:0_18:2	0.0013	**	0.78	↓
SM d32:2	0.0015	**	0.57	↓
PC 14:0_18:2	0.0016	**	0.58	↓
Cer d18:1_24:1	0.0016	**	1.43	↑
PC O-18:1_20:4	0.0018	**	1.15	↑
PC 17:0_18:2	0.0019	**	0.76	↓
LPE 18:1 sn-1	0.0021	**	0.67	↓
PC 18:1_18:2	0.0022	**	0.75	↓
PC 16:1_18:2	0.0023	**	0.67	↓
SM 34:0;O3	0.0024	**	0.7	↓
PI 18:1_20:4	0.0029	**	0.7	↓
PE O-20:1_18:2	0.0031	**	0.74	↓
PC 18:0_18:1	0.0035	**	0.75	↓
PE O-17:1_22:6	0.0043	**	0.6	↓
SM d33:1	0.0044	**	0.7	↓
PC 19:0_18:2	0.005	**	0.65	↓
PC 20:4_20:4	0.0057	**	0.72	↓
SM d24:2_14:0	0.0072	**	0.76	↓
PE O-18:3_20:4	0.0072	**	0.69	↓
SM d41:2	0.0078	**	0.49	↓
PE O-16:1_20:4	0.0091	**	0.73	↓
PI 16:0_18:2	0.0095	**	0.72	↓
Cer d43:2	0.011	*	1.58	↑
PE O-17:1_20:4	0.0114	*	0.69	↓
SM d36:3	0.0143	*	0.75	↓
PC 14:0_20:4	0.0148	*	0.62	↓
SM d18:1_16:1	0.0164	*	0.77	↓
PC 14:0_22:6	0.019	*	0.67	↓
PE O-18:2_20:4	0.0196	*	0.72	↓
PI 18:0_18:2	0.0223	*	0.78	↓
SM d34:0	0.0225	*	0.84	↓
LPE 22:5	0.0229	*	1.34	↑
LPE 20:4 sn-1	0.0289	*	0.77	↓
PE O-18:2_22:6	0.0344	*	0.76	↓
PE 18:1_18:2	0.0361	*	0.76	↓
PC 15:0_20:4	0.0374	*	0.77	↓
SM d40:1	0.0379	*	0.8	↓
SM d35:2	0.0405	*	0.81	↓
SM 34:1;O3	0.0418	*	0.84	↓
PE O-22:2_20:4	0.0418	*	1.37	↑
PC 17:1_18:2	0.0457	*	0.77	↓

**Table 3 cancers-15-03653-t003:** Significantly changed plasma lipids in female patients with DLBCL—data integration. n1, number of significant correlations in the DLBCL group; n2, number of significant correlations in the CTRL group; ns, non-significant. q-values represent BH corrected *p*-values. Mean lipid levels in DLBCL plasma are either increased (↑) or decreased (↓) compared to CTRL.

Lipid Species	VIP	q-Value	FC	Change Trend	n1	n2
S1P	2.113	0.00007	1.85	↑	0	2
SM 36:1	2.032	0.00140	2.10	↑	2	2
SM 34:1	1.938	0.00467	1.77	↑	1	2
PI 34:1	1.673	0.01131	0.62	↓	1	2
SM 42:1	1.423	ns	1.25	↑	1	2
PS 38:4	1.400	0.03456	1.49	↑	0	0
LPI 20:4	1.370	0.03456	1.45	↑	0	0
12(S)-HETE	1.180	0.02170	3.15	↑	0	2
LA	1.151	0.03325	1.46	↑	1	3
RvD1	1.098	ns	1.34	↑	0	0
PE 38:6	1.091	ns	1.34	↑	1	0
15(S)-HETE	1.025	0.08225	1.45	↑	0	1
TxB2	0.970	ns	3.12	↑	0	0
20-HETE	0.917	0.07350	1.45	↑	0	0
PE 38:4	0.905	ns	1.19	↑	0	2
5(S)-HETE	0.843	ns	1.30	↑	2	3
8(S)-HETE	0.522	ns	1.15	↑	2	3
AA	0.363	ns	1.13	↑	3	3

## Data Availability

The data presented in this study are available in this article, in the Appendix A. The raw datasets (mass spectra) generated and/or analysed during the study are available from the corresponding author Laura Bindila on reasonable request.

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
