# Peer review of "Disturbed Plasma Lipidomic Profiles in Females with Diffuse Large B-Cell Lymphoma: A Pilot Study"

_cancers, 2023, doi:10.3390/cancers15143653_

Round 1

Reviewer 1 Report

The manuscript is well written and the data are properly processed and statistically evaluated and finally interpreted. However, the main problem, in my opinion, is the analytical methodology, where only 35 lipids were selected for analysis and subsequent evaluation. Lipidomics is currently generally capable of semi-quantifying big hundreds of lipids in a single analysis and can offer detailed insight into pathobiochemistry. This is missing from the article and therefore I am not sure if this article is suitable for publication in this form.

Author Response

-The manuscript is well written and the data are properly processed and statistically evaluated and finally interpreted. However, the main problem, in my opinion, is the analytical methodology, where only 35 lipids were selected for analysis and subsequent evaluation. Lipidomics is currently generally capable of semi-quantifying big hundreds of lipids in a single analysis and can offer detailed insight into pathobiochemistry. This is missing from the article and therefore I am not sure if this article is suitable for publication in this form.

Answer: We have followed the reviewer’s suggestion and conducted untargeted plasma lipidomic profiling of DLBCL and control samples using modern, high-resolution 4D- lipidomics and relative quantification using class-specific internal standards (Lerner et al, Nat. Commun 2023). The untargeted data were subjected then to between-samples univariate statistical analysis, followed by principal component analysis (PCA) and orthogonal projections to latent structures discriminant analysis (OPLS-DA), much as we did for the 35 lipid species that were quantified by targeted analysis by LC MRM. The results of the untargeted analysis were incorporated into the revised manuscript, with new figures: PCA and OPLS-DA score plots, as well as PCA loading plots. The bulky results of the univariate statistical analysis we placed into the Supplementary files.

The reviewer mentioned detailed insight into pathobiochemistry through measurements of big hundreds of lipids from DLBCL plasma. However, comprehensive characterisation of the biochemical processes associated with the disease cannot be solely achieved from lipidomics data from patientsʼ plasmas. Such an objective would require a complete lipidome, metabolome, transcriptome, glycome etc landscape of all the cells involved in the tumour, that is all subsets of DLBCL cells, and all cells from the tumour microenvironment, including numerous immunocompetent cells, which was far beyond the scope of our study. Hence, we agree our study reveals a focused snapshot of the inflammatory, signaling and broad lipidome dysregulation in this cancer type.

Reviewer 2 Report

The novelty of this study lies in its investigation of the global plasma lipidome and sub-lipidome of inflammatory pathways in diffuse large B-cell lymphoma (DLBCL), which has not been reported previously. By focusing on female DLBCL patients and BMI-matched healthy controls, the study explores lipid mediators involved in inflammation, immunity, and cancer progression using targeted liquid chromatography coupled with multiple reaction monitoring.

The identification of eicosanoids as important factors in differentiating DLBCL and healthy status, particularly 12(S)-HETE, 15(S)-HETE, thromboxane B2, and lysophosphatidylinositol LPI 20:4, is a novel finding. This result implies significant changes in lipid metabolic and/or signaling pathways, specifically in the arachidonic acid lipoxygenase pathway and glycerophospholipid remodeling in the cell membrane.

Furthermore, the study suggests a set of lipids (S1P, SM 36:1, SM 34:1, and PI 34:1) as a potential DLBCL lipid signature, which is a new contribution to the field. The identification of this lipid signature could serve as a foundation for prospective validation in larger DLBCL cohorts and ultimately contribute to a better understanding of lipid dysregulations and pathways in this specific cancer type. Overall, the study provides valuable novel insights into the lipidomic profile of DLBCL and highlights potential lipid biomarkers that could be further explored for their diagnostic or therapeutic potential. I recommend its publication with minor modifications.

Line 72: please consider changing to “although novel treatments such as rituximab have improved…”

Table 1: Please consider moving it to “result” section. For clinical stage of DLBCL, the total number is 14 instead of 17. Please explain. Also, I am not aware there is stage V.

Table 1: Please consider providing clinical information (comorbidity such as diabetes, hypertension, hyperlipidemia), pathology diagnosis and molecular information on DLBCL patients, as they may have potential impact on signaling pathways: germinal center (GCB) vs. non-GCB, as well as expression of BCL-2, BCL-6, myc.

Line 421: Please correct the typo “,,”

Line 683: Please explain in this section that why only female patients with DLBCL were selected in this study.

The quality of English language in the manuscript is generally good. The text is well-written, clear, and easy to understand. However, there are a few instances where minor revisions could be made to improve clarity and readability. Here are some suggestions:

Line 391: "appropriate increments" could be rephrased to "relevant increases".

Line 425: Consider replacing "spread over" with "distributed across".

Line 429: Replace "non-separable" with "indistinguishable".

Line 461: Replace "n-6/n-3 PUFAs" with "n-6 and n-3 PUFAs".

Line 477: Consider changing "It should be emphasised" to "It should be noted".

Author Response

-The novelty of this study lies in its investigation of the global plasma lipidome and sub-lipidome of inflammatory pathways in diffuse large B-cell lymphoma (DLBCL), which has not been reported previously. By focusing on female DLBCL patients and BMI-matched healthy controls, the study explores lipid mediators involved in inflammation, immunity, and cancer progression using targeted liquid chromatography coupled with multiple reaction monitoring.

The identification of eicosanoids as important factors in differentiating DLBCL and healthy status, particularly 12(S)-HETE, 15(S)-HETE, thromboxane B2, and lysophosphatidylinositol LPI 20:4, is a novel finding. This result implies significant changes in lipid metabolic and/or signaling pathways, specifically in the arachidonic acid lipoxygenase pathway and glycerophospholipid remodeling in the cell membrane.

Furthermore, the study suggests a set of lipids (S1P, SM 36:1, SM 34:1, and PI 34:1) as a potential DLBCL lipid signature, which is a new contribution to the field. The identification of this lipid signature could serve as a foundation for prospective validation in larger DLBCL cohorts and ultimately contribute to a better understanding of lipid dysregulations and pathways in this specific cancer type. Overall, the study provides valuable novel insights into the lipidomic profile of DLBCL and highlights potential lipid biomarkers that could be further explored for their diagnostic or therapeutic potential. I recommend its publication with minor modifications.

Answer: We thank the reviewer for the positive appreciation and also for the helpful proofreading of the manuscript.

 We corrected all of the following:

-Line 72: please consider changing to “although novel treatments such as rituximab have improved…”

-Table 1: Please consider moving it to “result” section. For clinical stage of DLBCL, the total number is 14 instead of 17. Please explain. Also, I am not aware there is stage V.

Answer: We have moved the Table 1 to the Results as suggested.

Regarding the clinical stage of the patients, unfortunately it was our mistake, we copy-pasted the clinical stages as they were assessed at diagnosis, before the biopsy and histological examination of the tumour tissue. The three missing values were due to the uncertainty of the clinical stage in three tumours prior to the histological examination. The revised form of the manuscript now contains full data on the patients. Stage V is the end stage of lymphoma, when it turns into leukemia, the term often used by haematologists. We are thankful that the mistake was noticed.

-Table 1: Please consider providing clinical information (comorbidity such as diabetes, hypertension, hyperlipidemia), pathology diagnosis and molecular information on DLBCL patients, as they may have potential impact on signaling pathways: germinal center (GCB) vs. non-GCB, as well as expression of BCL-2, BCL-6, myc.

Answer: Regarding the expression of BCL-2, BCL-6 and MYC, we did assess them, and those patients with detected rearrangements in these three oncogenes were excluded from the study as high-grade lymphomas. Due to a humble number of patients, we tried to uniform the DLBCL group by recruiting only treatment-naïve women, only those without previous cancers, without comorbidities, that were in normal range of clinical lipids, that were not overweight, clinical stage III or IV and without high-grade lymphomas (meaning no rearrangements in MYC, bcl-2 or bcl-6). We detailed this in the revised manuscript. Maybe it is worth noting in here that Lupino and collaborators (2019) measured the elevated levels of sphingosine kinase 1, enzyme responsible for S1P synthesis in both ABC- and GCB- DLBCL subtypes (doi:10.1038/s41375-019-0478-9), which goes along with our thoughts that elevations in lipid species that had a great statistical significance in comparisons to controls overwrote the differences in DLBCL subtypes.

-Line 421: Please correct the typo “,,”

-Line 683: Please explain in this section that why only female patients with DLBCL were selected in this study.

Answer: The answer to this issue is the most trivial but males consistently refused to sign the Informed consent to participate in the study, whereas female patients were much more interested to learn about the disease and participate. Secondly, knowing that DLBCL hits more men than women and that men have been (until recently) much more often recruited in cancer studies we thought that we should conduct our study on female patients. Finally, in a comparable plasma lipidomics study done on a haematological cancer AML in 16 males and 4 females (Pabst et al, 2017; DOI: 10.1016/j.bbacli.2017.03.002), the authors stated: ,, A total of 50 lipid molecules from several classes were found to be highly statistically significantly altered in the plasma of AML patients relative to healthy controls. Selected lipids were tested for gender differences. No statistically significant differences between males and females, neither for the AML cases nor the controls, were found for these plasma lipids. Hence, the metabolic rewiring displays no gender differences.ˮ

The quality of English language in the manuscript is generally good. The text is well-written, clear, and easy to understand. However, there are a few instances where minor revisions could be made to improve clarity and readability. Here are some suggestions:

-Line 391: "appropriate increments" could be rephrased to "relevant increases".

-Line 425: Consider replacing "spread over" with "distributed across".

-Line 429: Replace "non-separable" with "indistinguishable".

-Line 461: Replace "n-6/n-3 PUFAs" with "n-6 and n-3 PUFAs".

-Line 477: Consider changing "It should be emphasised" to "It should be noted".

Answer: We have followed the reviewer’s suggestion and reworded the text accordingly.

Reviewer 3 Report

English is good but it is noteworthy that introduction is made of many separate paragraphs, an integration to make it more fluent and easier to follow would be appreciated. The same applies also to discussion that is a also hypertrophic, but do not present integrated views of the results or consider previous works such as Tiantian Am J Cancer Res. 2023; 13(2): 475–484 or Wang Ann Hematol. 2023 Feb;102(2):393-402.

Please include healthy controls in table 1 (in addition to reference values) and specify if also them are recruited in Belgrade in the same period. Only women were recruited but this choice is presented only in the discussion. An age difference is randomly presented, differences in total plasma lipid levels?

Please briefly clarify the extraction procedure used, one protocol is briefly described but in the text are cited a Lerner 2017 and a modified version (ref.61).

Many statistics (redundant?) were applied, if this is a pilot study to determine the data analysis pipeline disclose it an illustrate the selected pipeline.

Some redundancy and tendency to write many independent paragraphs more than a text.

Author Response

-Minor editing of English language required. Some redundancy and tendency to write many independent paragraphs more than a text.

Аnswer: We thank the reviewer for a thorough review of our manuscript. Indeed, the original version of the manuscript was lengthy, somewhat fragmented and there were redundancies in Introduction and Discussion. We thoroughly revised the text to improve readability and clarity, concision and presentation of the goals and study design and deleted redundancies. Practically, we re-wrote the Introduction and Discussion.

The Discussion section was also thoroughly re-structured to integrate our main findings into a well-integrated story on altered lipidome in DLBCL. We removed all the reference and discussion on solid cancers in relation to our results, leaving only haematological cancers for the comparisons.

-English is good but it is noteworthy that introduction is made of many separate paragraphs, an integration to make it more fluent and easier to follow would be appreciated. The same applies also to discussion that is also hypertrophic, but does not present integrated views of the results or consider previous works such as Tiantian Am J Cancer Res. 2023; 13(2): 475–484 or Wang Ann Hematol. 2023 Feb;102(2):393-402.

Аnswer: The first part of the reviewerʼs comment has been already addressed above. The authors appreciate the constructive criticism of clarity of writing and presentation. Indeed, it helped us revise both the text and extraction of the essential findings to present a more integrate view of the results. We have added the recommended two articles by Yu et al, and Wang et al (2023).

-Please include healthy controls in table 1 (in addition to reference values) and specify if also them are recruited in Belgrade in the same period. Only women were recruited but this choice is presented only in the discussion.

Аnswer: The authors made corrections as suggested. The controls were recruited from the two research centres at the University of Belgrade, in March 2019, and they came to give blood samples and filled in the Informed consent at the Clinical Hospital Centre Zemun-Belgrade. We have added this info in the revised manuscript. Only women were recruited since many male patients with DLBCL refused to give a written consent to participate in our study. We included information that the study was performed with plasma samples from females in the Introduction also. The reviewer is referred to the Simple Summary, Abstract and Title of the original manuscript, where the specification of “females” was included.

-An age difference is randomly presented, differences in total plasma lipid levels?

Аnswer: Thank you for pointing this out. Indeed, there was an age difference between the two cohorts, DLBC (median 59, range 36-69) and CTRL (median 49, range 30-64). Regarding this issue, we must make two points: i) it is close to impossible to find older healthy females without comorbidities, dyslipidaemia, who do not take hormone replacement therapy, or NSAIDs or oil-based supplements; ii) the age gap in case-control studies on haematological cancers is not a rarity, see also: Pabst et al (2017) The plasma lipidome in acute myeloid leukemia at diagnosis in relation to clinical disease features, page 112: ,, It is interesting to note that, although the AML patients were significantly older, with more males, than the blood donor controls recruited, they, counterintuitively, had lower plasma lipids…ˮ. Our DLBCL cohort had mean values of triglycerides, cholesterol and liver enzymes within the normal range (Table 1), therefore, we did not anticipate these factors to affect the lipidome in our study.

-Please briefly clarify the extraction procedure used, one protocol is briefly described but in the text are cited a Lerner 2017 and a modified version (ref.61).

Аnswer: Thank you for noticing this. We decided to shorten the description of the lipid extraction procedure, since it has been published in details, in several research and (video)protocol papers by the co-authors at Clinical Lipidomics Centre Mainz. Instead of the extensive description of the lipid extraction procedure, we cited the relevant papers. We added brief description of the methods of extraction and analysis of the untargeted lipidomics and included reference to the study.

-Many statistics (redundant?) were applied, if this is a pilot study to determine the data analysis pipeline disclose it and illustrate the selected pipeline.

Answer: We do agree with the reviewer that many statistics were applied, so we re-examined the need for so many (multivariate) analyses. The apparent redundancy arose more from the results of these analyses, than from the way of analysing the dataset, because they all point out to changes in levels of same four lipid species: S1P, SM 34:1, SM 36:1 and PI 34:1 that were the signature of DLBCL plasma. To follow the pattern of other lipidomic studies, we left the results of univariate (classic) statistics with between-groups comparisons, unsupervised PCA because it can reveal clustering within the group, and OPLS-DA as a standard tool that comes along with such analyses, revealing variables of importance in distinguishing cancer cases from controls, since these variables are hints for future studies of potential biomarkers. We removed Principal component regression, since it emphasized the major role of S1P, PI 34:1, SM 36:1 and SM 34:1 in discriminating healthy state from DLBCL, the same four lipid species that were already shown by the univariate statistics and OPLS-DA obtained VIP values to be the major lipid signatures of the DLBCL. The Volcano plot in Fig. 6 was deleted for the same reason. We have also removed Fig. 5, showing the loadings and VIP values obtained through OPLS-DA to the Supplemental file since Table 6. displays these VIP values.

We intended to confirm, using multiple data analysis tools on the same dataset, the findings and concurrently compensate for lower than wished for nr of participants with solid data. We removed as per suggestion, redundant analyses and shortened accordingly the statistics sections in Materials and Methods and description throughout the manuscript.

Reviewer 4 Report

Dysregulation of the lipidome is a characteristic of cancer and inflammation. However, the global plasma lipidome and sub-lipidome of inflammatory pathways have not been previously investigated in diffuse large B-cell lymphoma (DLBCL). To address this gap, a pilot study was conducted to quantify lipid mediators involved in inflammation and immunity, as well as those implicated in cancer progression, using targeted liquid chromatography coupled with multiple reaction monitoring. The study compared plasma lipid variation in female DLBCL patients with BMI-matched healthy controls. The results indicated notable changes in lipid metabolic and signaling pathways, particularly in arachidonic acid (AA) lipoxygenase pathway and glycerophospholipid remodeling in the cell membrane. The study identified a set of lipids as the DLBCL lipid signature, including S1P, SM 36:1, SM 34:1, and PI 34:1, which could serve as a basis for the prospective validation in larger DLBCL cohorts. The plasma lipid profiling of DLBCL patients provides a better understanding of the specific lipid dysregulations and pathways in this cancer type. Specifically, the study highlights the significance of eicosanoids, such as 12(S)-HETE, 15(S)-HETE, thromboxane B2, and lysophosphatidylinositol LPI 20:4, in the separation of DLBCL and healthy status. Here are some potential limitations of the study: Small sample size - The study only included 38 total participants, with 17 DLBCL patients. Larger studies would be needed to validate and generalize the findings. Limited demographic information - Only basic demographic data were provided for participants. Information on other factors like smoking status, medications and diet could impact lipid levels and confound comparisons. Cross-sectional design - The study had a cross-sectional design, measuring lipid levels at one time point. This limits inferences about changes in lipids over disease progression. Longitudinal studies would be needed. Limited clinical information - Minimal clinical details were provided on the DLBCL patients, aside from age, sex and disease stage. Other factors like treatment history, tumor characteristics and comorbidities could influence lipid levels. Potential selection bias - The exclusion criteria meant that only certain DLBCL patients were included, which may not represent the full population. This could introduce selection bias. Limited lipid profiling - The study focused on a subset of lipids, mainly eicosanoids, phospholipids and fatty acids. A broader lipidomic analysis could provide more insights. Lack of mechanistic insights - The study identified correlational changes in lipids but could not determine the mechanisms driving these alterations or their functional significance. Further studies would be needed. In summary, while the study identifies some interesting differences in plasma lipids between DLBCL patients and controls, the limitations highlight areas for improvement in future research to validate and expand upon these initial findings. Larger, longitudinal studies with broader lipid profiling and mechanistic investigations would help advance this work. In the analysis of DLBCL, it is important to examine the role of various cellular components of the tumor microenvironment, such as mast cells, macrophages, and lymphocytes. Several approaches have been explored to confront available evidence, revealing three key points about DLBCL tumor progression. This reviewer personally misses some insights within the introduction and discussion sections regarding novel aspects of DLBCL. Evaluating the tumor microenvironment is crucial for understanding the progression of DLBCL. In order to comprehend the tumor progression of DLBCL, various components of the tumor microenvironment, such as mast cells, macrophages, and lymphocytes, have been thoroughly examined. We explored multiple approaches to address the existing evidence, resulting in three key observations. DLBCL is characterized by the proliferation and accumulation of malignant B cells, both within lymph nodes and outside of them. The subsequent establishment of a conducive environment for cancer becomes crucial in patients with both lymph node and extranodal manifestations. DLBCL cells interact with mature stromal cells and blood vessels, providing protection to the tumor and inhibiting immune responses, while ensuring the provision of nutrients and oxygen. Additionally, individual cells may find refuge in protected niches within both nodal and extranodal sites, acting as a potential source of residual disease and relapse. Please refer to PMID: 32664527 and expand.

Minor typos.

Author Response

-Dysregulation of the lipidome is a characteristic of cancer and inflammation. However, the global plasma lipidome and sub-lipidome of inflammatory pathways have not been previously investigated in diffuse large B-cell lymphoma (DLBCL). To address this gap, a pilot study was conducted to quantify lipid mediators involved in inflammation and immunity, as well as those implicated in cancer progression, using targeted liquid chromatography coupled with multiple reaction monitoring. The study compared plasma lipid variation in female DLBCL patients with BMI-matched healthy controls. The results indicated notable changes in lipid metabolic and signaling pathways, particularly in arachidonic acid (AA) lipoxygenase pathway and glycerophospholipid remodeling in the cell membrane. The study identified a set of lipids as the DLBCL lipid signature, including S1P, SM 36:1, SM 34:1, and PI 34:1, which could serve as a basis for the prospective validation in larger DLBCL cohorts. The plasma lipid profiling of DLBCL patients provides a better understanding of the specific lipid dysregulations and pathways in this cancer type. Specifically, the study highlights the significance of eicosanoids, such as 12(S)-HETE, 15(S)-HETE, thromboxane B2, and lysophosphatidylinositol LPI 20:4, in the separation of DLBCL and healthy status. Here are some potential limitations of the study: Small sample size - The study only included 38 total participants, with 17 DLBCL patients.

Answer: The reviewer is right in pointing that this is indeed only a pilot study. Serbia is a small country and DLBCL is relatively rare tumour that affects more males than females. We designed the study to recruit all females that presented to the Department of Haematology of Clinical Hospital Centre Zemun, Belgrade, from May 2017. to September 2018, consecutively, but with high-grade DLBCL (stage III and IV). Since we wanted only patients without previous malignancies, concurrent malignancies, severe chronic diseases, evidence of cancer cachexia or a history of medications/treatments that affect lipid metabolism such as taking oral contraceptives, omega-3 supplements or non-steroidal anti-inflammatory drugs (NSAID), the number of patients in the study was 17. Our work is comparable with the work of Pabst et al.,( Pabst, T.; Kortz, L.; Fiedler, G.M.; Ceglarek, U.; Idle, J.R.; Beyoglu, D. The plasma lipidome in acute myeloid leukemia at diagnosis in relation to clinical disease features. BBA Clin 2017, 7, 105-114, doi:10.1016/j.bbacli.2017.03.002), where the authors carried out an untargeted lipidomic investigation of plasma from only 20 AML patients and 20 controls using UPLC-ESI-QTOFMS.

We are confident that our study, after including the comprehensive plasma lipidome analysis using high resolution untargeted 4D lipidomics of DLBCL and control cohort in the revised manuscript, despite the small sample size, offers a valuable insight into picture of DLBCL, and this is not only because it is the first global lipidomic profiling of plasma in DLBCL. Moreover, we displayed a rare view on metabolites of the arachidonic acid pathway and their precursors in this immune system cancer.

-Larger studies would be needed to validate and generalize the findings.

Answer: Indeed, we are keen on prospective validation and complementing of these findings in larger cohorts.  The effort of collecting more samples from males and females with DLBCL is significant and require many years to reach large cohorts in a particular region such as in Serbia. The need for the validation of omics (and lipidomics data) is widely recognised, and leaves space for many studies engaging different sexes, ethnicities, nutrition etc etc. We did not inflate the value of our study, we presented the results within the limits of the present study design and size, and did not make any claim pertaining to biomarkers discovery. We focused rather our study on the inflammatory arm of DLBCL in female patients, particularly arachidonic acid pathway, the latter we followed throughout, from its phospholipid precursors to its metabolites. Furthermore, eiCs are relatively rarely measured in lipidomic studies in human samples, and even more rare in combination to global lipidomics despite their very significant role in the pro-cancer and pro-inflammatory signalling, and this was the major value of our study (besides the untargeted lipidomics).

-Limited demographic information - Only basic demographic data were provided for participants. Information on other factors like smoking status, medications and diet could impact lipid levels and confound comparisons.

Answer: We do agree with the reviewer that all the mentioned factors could have affected the measured lipid levels. Regarding the smoking status, female patients with DLBCL were smokers, non-smokers, ex-smokers, as were the control females, but we must emphasize few points: a) the cohorts described in this manuscript were of insufficient size for the stratification of subjects. Fortunately, the PCA score plot did not reveal any clusters inside the studied DLBCL cohort (nor within the CTRL cohort), which meant there were no subsets of datasets.

In order to preclude the potential effect of (lipid-altering)medications on the plasma lipidome signature, we excluded all DLBCL patients that had a history of medications/treatments that affect lipid metabolism such as omega-3 supplements or non-steroidal anti-inflammatory drugs (NSAID), just as we did when we chose control females. Therefore, at least the impact of one factor, medications, on the lipidome was excluded.

Few studies showed that lipids in plasma are genetically determined and tightly controlled and not all of them display diurnal variations or great changes during the course of time. Besides, diurnal changes are mitigated, similar to current clinical blood lipid tests, by standardized and appropriate sampling protocol (like fasting,) and sample preprocessing protocols. In fact, lipidomics has manage to render implementation in praxis of new clinical markers, which attests its suitability for clinical and preclinical discoveries despite life style variability. 

The fact that our DLBCL female patients had triglyceride and total cholesterol values within the normal range is in this context also a supporting argument.

DLBCL is a highly heterogeneous cancer, both genetically and metabolically, with many subsets of cancer cells that escape the grip of research designs. But still, some lipid species such as S1P, its precursor SMs, then PI 34:1 and LPI 20:4 were profoundly and consistently altered in almost all DLBCL plasma samples, regardless of their diet, or whether these samples were from smokers, non-smokers, ex-smokers. Importantly, though our patients were in the cancer stages III and IV, and their subtype was either GCB-like or non-GCB like, the levels of above-mentioned four major lipid species were significantly altered, not showing differences based on the cancer stage or subtype. Accordingly, we don’t expect that the smoking status or diet would change our major findings. What we accordingly highlight here is that the change in the plasma lipidome is obviously directed by the cancer itself and is significant enough to overwrite the interindividual differences within the DLBCL cohort. Hence we consider these lipids as sound candidates for lipid signatures of this disease.

-Cross-sectional design - The study had a cross-sectional design, measuring lipid levels at one time point. This limits inferences about changes in lipids over disease progression. Longitudinal studies would be needed.

Answer: In the realm of cancer lipidomics, as far as we know, the longitudinal studies are still rare. In the case of DLBCL, the incidence is 7 cases per 100 000 people per year and longitudinal sampling with significant cohort sizes is a time and logistic demanding. The longitudinal study falls beyond the scope of this project. However, we are confident that our present study, though limited in few aspects, provide a sound basis to inform and focus future lipidomic and omic studies in general of cancer tissue and patientsʼ plasma.on this highly diverse cancer.  

Limited clinical information - Minimal clinical details were provided on the DLBCL patients, aside from age, sex and disease stage. Other factors like treatment history, tumor characteristics and comorbidities could influence lipid levels.

Answer: Thank you for pointing this out. We actually missed to give a complete information on our DLBCL patients upon the submission, which was a mistake created due to the wrongful copy-paste process of the text and Table with patient data in Materials and methods. When the patients presented for the first time at Clinical Hospital Centre Zemun-Belgrade, they were diagnosed as having cancer stage I to V based on their clinical biochemical analyses, before the biopsy. However, the histological diagnosis was made using the lymph node biopsy or biopsy of the primary extranodal site, so the patients were classified according to the exact tumour stage and the subtype: GCB-like and non-GCB-like DLBCL. This information is now included in the revised manuscript. Our female patients were all either stage III or stage IV, since we wanted to randomise the study, given that we had a relatively small number of subjects. Of note, stages III and IV are now considered a single category because they have the same treatment and prognosis, which is why we did not stratify the patients into two groups (DOI: 10.1200/JCO.2013.54.8800).

Regarding the treatment history, there was not any, all patients were treatment-naïve, this info was/is included in the Materials and methods. Tumour characteristics are also added: ,,There were eight patients with GCB-like subtype and nine patients with non-GCB-like subtype of DLBCL. There were no patients with the high-grade B-cell lymphomas in the studyˮ. There were also no comorbidities, as included in the revised manuscript: ,, All patients with previous malignancies, concurrent malignancies, severe chronic diseases, evidence of cancer cachexia or a history of medications/treatments that affect lipid metabolism such as omega-3 supplements or non-steroidal anti-inflammatory drugs (NSAID) were excluded from the study.ˮ

-Potential selection bias - The exclusion criteria meant that only certain DLBCL patients were included, which may not represent the full population. This could introduce selection bias.

Answer: This is indeed an important point, but we do think it is related to the issue of the sample size we discussed above. At first, we intended to recruit all treatment-naïve DLBCL patients that presented to the Department of Haematology at Clinical Hospital Centre Zemun during the six months period dedicated to collect the samples. We emphasize here that the majority of the newly diagnosed patients actually were in the advanced stage of the disease, since the early DLBCL often stays unnoticed by the patients (as in https://doi.org/10.1056%2FNEJMra2027612. Because we didn’t have enough DLBCL patients with the limited stage disease (clinical stages I and II), we recruited only advanced stage cancers (clinical stages III and IV) in order to be able to observe the lipidomic changes. Finally, for some reason, females were much more ready to talk with the haematologist about the disease and to sign the informed written consent to participate in the study, while men mostly refused to do so. So, we recruited consecutive females with stage III and IV DLBCL.

Certainly, we exclude the patients with other types of cancers, to better ascertain the origins of the altered lipidome. The same stands for the comorbidities. Regarding therapies, the rituximab or some novel combinations of drugs used for the treatment of DLBCL may affect the lipid metabolism so we excluded the patients on the therapy or after the therapy.

Additional note on the selection bias: we would be concerned about the potential bias if the measured values for the four major lipid signatures partially overlapped with the mean values for controls, with the p-value for between-groups comparisons close to 0.05. Since we detected a profoundly altered lipid species in the majority of DLBCL samples, we are pretty certain it is a general signature of this cancer.

-Limited lipid profiling - The study focused on a subset of lipids, mainly eicosanoids, phospholipids and fatty acids. A broader lipidomic analysis could provide more insights.

Answer: This is also an issue raised by the other reviewer. In order to address this concern of two reviewers, we conducted the untargeted lipidomic investigation of our DLBCL cohort and the control cohort. Our initial aim was to study arachidonic acid, its main metabolites and phospholipid precursors and few pro-cancer and pro-inflammatory lipid mediators to get a glimpse of the inflammatory and immune-mechanism mediated by lipids and their footprint in  plasma of patients. Adding the untargeted lipidomics confirmed and complemented the findings from the targeted study. This combinatorial, focused approach will better inform prospective larger scale DLBCL cohort studies in terms of both hypothesis-based design and data mining. Prospectively, certainly, in largere cohorts we could better stratify patients based on sex, DLBCL metabolic subsets, DLBCL clinical grade and extract more biological knowledge. However, such a study is beyond our scope this time.

-Lack of mechanistic insights - The study identified correlational changes in lipids but could not determine the mechanisms driving these alterations or their functional significance. Further studies would be needed.

Answer: The reviewer is right and answer to this concern is a complex one. As the reviewer pointed out in order to comprehend the tumour progression of DLBCL, various components of the tumour microenvironment, such as mast cells, macrophages, and lymphocytes, must be thoroughly examined. Deep mechanistic insights can be revealed by system medicine, system biology approaches with integrated lipidonics or multiple omics, but this has been beyond the scope of this study. 

However, it is worthwhile  to mention here, that investigation of lipid mediators or inflammation and immunity pertaining to specific synthetic pathways are rarer investigated in human biospecimens than global lipidomics for example, and even rarer in combination. Therefore, we aimed to have a focused study designed that would reveal more mechanistic aspects than just global lipidome changes which is, in general expected by the sheer acquisition of cancer cells of new metabolic statuses to proliferate and survive. Accordingly, we hope to offer a good bases for focused and informed systems biology and medicine studies of DLBCL cancer in the future.  

 Multiple omics data though would help us reach some extent of the mechanistic explanation of the lipidome changes, but still we could not guarantee the success knowing that so many genetic and metabolic subtypes and subsets of DLBCL exist and that many different cancer cells may exist even in the same patient. As the reviewer accentuated, further studies are needed, which will be paving the way for development of more personalized approaches to treatment. The answer is linked to the answer bellow.

-In summary, while the study identifies some interesting differences in plasma lipids between DLBCL patients and controls, the limitations highlight areas for improvement in future research to validate and expand upon these initial findings. Larger, longitudinal studies with broader lipid profiling and mechanistic investigations would help advance this work.

-In the analysis of DLBCL, it is important to examine the role of various cellular components of the tumor microenvironment, such as mast cells, macrophages, and lymphocytes.

Answer: The reviewer raised an essential point, to make a real progress in revealing the mechanisms of DLBCL, each cell type involved in the carcinogenesis must be studied independently. In addition to the answer above, for this to work, also cellular multi omics must be employed where the normal development and disease processes can be studied and dissected at a single-cell population resolution. Unfortunately,  such inference falls beyond the scope of this project.

-Several approaches have been explored to confront available evidence, revealing three key points about DLBCL tumor progression. This reviewer personally misses some insights within the introduction and discussion sections regarding novel aspects of DLBCL. Evaluating the tumor microenvironment is crucial for understanding the progression of DLBCL. In order to comprehend the tumor progression of DLBCL, various components of the tumor microenvironment, such as mast cells, macrophages, and lymphocytes, have been thoroughly examined. We explored multiple approaches to address the existing evidence, resulting in three key observations. DLBCL is characterized by the proliferation and accumulation of malignant B cells, both within lymph nodes and outside of them. The subsequent establishment of a conducive environment for cancer becomes crucial in patients with both lymph node and extranodal manifestations. DLBCL cells interact with mature stromal cells and blood vessels, providing protection to the tumor and inhibiting immune responses, while ensuring the provision of nutrients and oxygen. Additionally, individual cells may find refuge in protected niches within both nodal and extranodal sites, acting as a potential source of residual disease and relapse. Please refer to PMID: 32664527 and expand.

Answer: The authors appreciate this point made by the reviewer, and we added several novel references on DLBCL, including the recommended one under PMID 32664527, and discussed  them and their findings in details, which are relevant to the lipidomic study described herein.

-This reviewer personally misses some insights within the introduction and discussion sections regarding novel aspects of DLBCL.

Answer: We studied numerous new works on DLBCL, but these most often dealt with a plethora of novel genetic subsets of this tumour, proposals of new taxonomies, gene rearrangements, mutations or deletions, activation of transcription factors, up- and down-regulation of proteins involved in cell fate and apoptosis, cell-to-cell variability in the tumour mass, the molecular abundances of signalling proteins such as BCL-2-family, biological functions of epigenetic alterations in DLBC, with a great number of the studies being performed on cell lines or tumour tissue. Numerous articles dealt with the patientsʼ response to anti-cancer therapy.

However, there were no articles on metabolic or lipidome changes in this tumour. For the sake of shortness and to avoid hypertrofic Introduction or Discussion (as the third Reviewer pointed out) we had to keep referring to metabolic/lipidomic changes in haematological cancers particularly lymphomas. But we included the new articles into the revised manuscript such as:

1) Pang et al (2023) Metabolic Reprogramming and Potential Therapeutic Targets in Lymphoma. Int J Mol Sci

2) Yu et al (2023) Prognostic and therapeutic value of serum lipids and a new IPI score system based on apolipoprotein A-I in diffuse large B-cell lymphoma. Am J Cancer Res

5) Wang et al (2023) Altered serum lipid levels are associated with prognosis of diffuse large B cell lymphoma and influenced by utility of rituximab. Ann Hematol

6) Cutmore (2023) Genetic Profiling in Diffuse Large B-Cell Lymphoma: The Promise and the Challenge. Mod Path

8) Kapadia et al (2018) Fatty Acid Synthase induced S6Kinase facilitates USP11-eIF4B complex formation for sustained oncogenic translation in DLBCL. Nat Commun

9) Eberlin et al (2014) Alteration of the lipid profile in lymphomas induced by MYC overexpression. PNAS

10) Lu et al (2023) COX-2/PGE2 upregulation contributes to the chromosome 17p-deleted lymphoma

Round 2

Reviewer 4 Report

The authors have clarified several of the questions I raised in my previous review. Most of the major problems have been addressed by this revision.